# Do Concept Bottleneck Models Respect Localities?

**Naveen Raman**                                                      *naveenr@cmu.edu*
*Carnegie Mellon University*

**Mateo Espinosa Zarlenga**                                           *me466@cma.ac.uk*
*University of Cambridge*

**Juyeon Heo**                                                        *jh2324@cam.ac.uk*
*University of Cambridge*

**Mateja Jamnik**                                           *mateja.jamnik@cl.cam.ac.uk*
*University of Cambridge*

**Reviewed on OpenReview:** *https://openreview.net/forum?id=4mCkRbUXOf*

## Abstract

Concept-based explainability methods use human-understandable intermediaries to produce explanations for machine learning models. These methods assume concept predictions can help understand a model's internal reasoning. In this work, we assess the degree to which such an assumption is true by analyzing whether concept predictors leverage "relevant" features to make predictions, a term we call *locality*. Concept-based models that fail to respect localities also fail to be explainable because concept predictions are based on spurious features, making the interpretation of the concept predictions vacuous. To assess whether concept-based models respect localities, we construct and use three metrics to characterize when models respect localities, complementing our analysis with theoretical results. Each of our metrics captures a different notion of perturbation and assess whether perturbing "irrelevant" features impacts the predictions made by a concept predictors. We find that many concept-based models used in practice fail to respect localities because concept predictors cannot always clearly distinguish distinct concepts. Based on these findings, we propose suggestions for alleviating this issue.

## 1 Introduction

Concept-based explainability provides insights into black-box machine learning models by constructing explanations via human-understandable "*concepts*" (Kim et al., 2018; Ghorbani et al., 2019; Koh et al., 2020). For example, we can explain that an image was predicted to be a cherry because we predicted that the concepts "Red Fruit" and "Circular Fruit" were present in the image. Within this paradigm, Concept Bottleneck Models (CBMs) (Koh et al., 2020) are a family of models that first predict concepts from an input (via a *concept predictor*) and then predict a downstream label from these concepts (via a *label predictor*). This design allows CBMs to provide concept-based explanations via their predicted concepts (Chauhan et al., 2022; Shin et al., 2023; Espinosa Zarlenga et al., 2023b).

Models within the concept-based paradigm rely on the assumption that a model's concept predictions serve as an explanation for its predictions (Koh et al., 2020). For example, if the concept predictor for "Red Fruit" mistakenly learns to predict the "Circular Fruit" concept instead, then explanations based on the "Red Fruit" concept predictor would be incorrect as this predictor fails to reflect the presence of "Red Fruit." Therefore, the faithfulness of concept predictors to their underlying concepts is critical because these models operate under the strong assumption that concept predictions align with their corresponding semantic concepts.

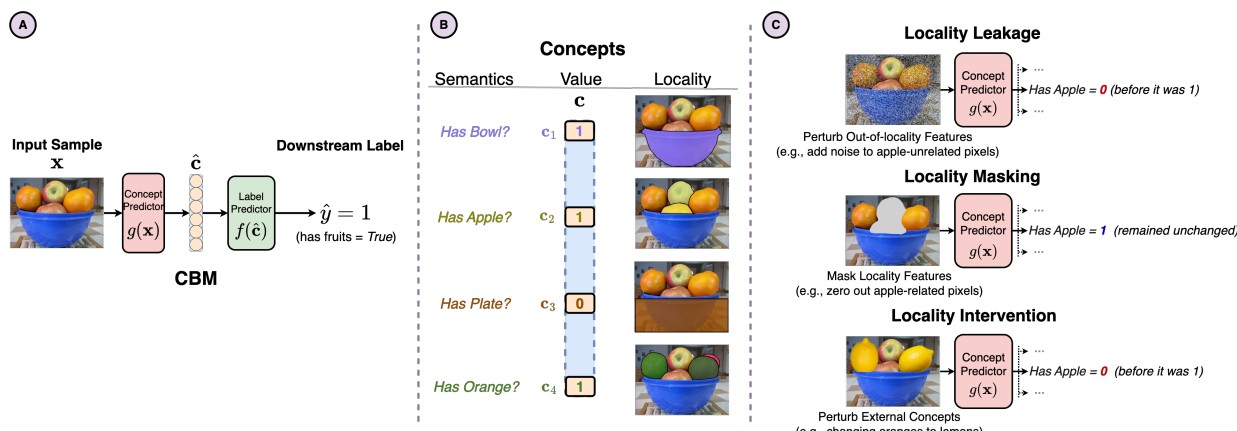

Figure 1: We investigate whether concept-based models properly capture known concept "localities". For example, we train a CBM to predict whether an image has a fruit (left panel A), with concepts such as "Has Bowl" and "Has Apple". Here, concepts are localised to image subregions (middle panel B). We study whether CBMs learn these localities by analysing how changes to input features, both within and outside a concept's locality, impact a CBM's predictions (right panel C).

Our work analyzes the faithfulness of concept predictors by assessing the degree to which a model's concept predictions are a proxy for a model's internal reasoning. We do so to understand whether explanations from concept-based models are *trustworthy*, that is whether we can rely on the explanations from concept-based models to gain insights (Vashney, 2022). We analyze the relationship between concept predictors and a model's internal reasoning through *locality*, which refers to the idea that oftentimes only a subset of features are needed to predict a concept (Figure 1). For example, we assess the degree to which the prediction of the concept "Red Fruit" is influenced by background features irrelevant to that concept. We construct a set of experiments to assess whether concept-based models respect locality, as this can give us insight into their trustworthiness. Our results reveal that concept-based models often fail to respect localities, a significant shortcoming that casts doubt on their interpretability.

Our contributions are as follows: we (1) construct three metrics to quantify whether models respect localities found in datasets, (2) conduct thorough experiments across a variety of architectural and training choices to characterize when models respect localities, and (3) propose theoretical models to better understand the impact of dataset construction on locality. [1]

## 2   Related Works

**Explainability and Locality**   Our work is broadly situated within the eXplainable Artificial Intelligence (XAI) field, which aims to elucidate the reasons behind a black-box model's predictions. Developing XAI methods is difficult because of the variety of desiderata for XAI methods (Rudin et al., 2022), including trustworthiness (Shen, 2022) and accuracy (Lipton, 2018). While a variety of XAI methods are used in practice, such as saliency maps (Baehrens et al., 2010) and SHAP (Lundberg & Lee, 2017), prior work has demonstrated the fragility of those methods by analyzing the patterns these methods learn (Adebayo et al., 2018). Our work follows a similar perspective by studying the fragility of concept-based explanations through the lens of locality.

**Concept-based Explainability**   Our investigation into locality touches upon various other phenomena in concept-based models. Concept-based models use human-understandable concepts to explain a machine learning model's prediction (Kim et al., 2018). Methods vary from fully concept-supervised (Koh et al., 2020; Chen et al., 2020; Yuksekgonul et al., 2022; Kim et al., 2023), where one has access to task and concept labels, to concept-unsupervised (Ghorbani et al., 2019; Yeh et al., 2020; Oikarinen et al., 2023; Pham et al.,

---

[1]We include our code and dataset construction details here: `https://github.com/naveenr414/Spurious-Concepts`

2024), where only task labels are available. In this paper, we explore CBMs (Koh et al., 2020) due to their ubiquity as building blocks for state-of-the-art methods (Espinosa Zarlenga et al., 2022; Yuksekgonul et al., 2022; Kim et al., 2023; Oikarinen et al., 2023; Espinosa Zarlenga et al., 2023c).

Prior work in concept-based explainability has shown that concept-based models are prone to "*concept leakage*", where label-specific information is undesirably encoded into concept predictions (Mahinpei et al., 2021; Havasi et al., 2022). Follow-up works studied how leakage may be (1) detected at inference (Marconato et al., 2022), (2) measured at test-time (Espinosa Zarlenga et al., 2023a; Marconato et al., 2024), (3) affected by inter-concept correlations (Heidemann et al., 2023), and (4) exploited for adversarial attacks (Sinha et al., 2023). Our work builds on these by studying the related but separate phenomena of locality, which captures the robustness of concept predictors to perturbations. We build on similar work on spurious concept correlations (Margeloiu et al., 2021; Furby et al., 2023; 2024) by introducing metrics to rigorously understand the origin of this phenomenon in different architectural choices. Our work additionally builds on literature studying inter-concept relationships in concept-based models by analyzing how incorrectly learning such relationships can decrease the robustness of concept predictors (Raman et al., 2024). Finally, related to our work is prior work which develops metrics for understanding concepts (Espinosa Zarlenga et al., 2023a; Marconato et al., 2024); however, in contrast to these works, we focus on quantifying the ability of concept predictors to rely on the relevant sets of features, rather than capturing whether they encode impurities or leakages. We include a further comparison of our work with other metrics in Appendix B.7.

**Spurious Correlations and Locality**   Our work investigates whether concept predictors rely only on relevant features while ignoring irrelevant features. Similar to our emphasis on whether concept-based models rely on irrelevant features, prior work has investigated whether deep learning models exploit spurious correlations (Geirhos et al., 2020; Ye et al., 2024). Previous work in spurious correlations investigated how factors such as model size (Sagawa et al., 2020), loss function (Levy et al., 2020), and data augmentation (Srivastava et al., 2020) impact spurious correlations. In this work, we similarly aim to systematically understand which factors impact localities in concept-based models. However, we differentiate ourselves from prior studies of spurious correlations because we (a) demonstrate that locality may fail to hold even in simple situations, and (b) study how a failure of locality can lead to the breakdown of interpretability for popular concept-based methods.

# 3   Why do we need locality?

We first formally define the problem of concept-based learning, then explain why we can use locality to understand the relationship between model predictions and model reasoning.

## 3.1   Defining Concept-Based Learning

Concept-based learning is a supervised learning task where, given a set of input features, models produce a label prediction and a corresponding concept-based explanation. In concept-based learning we are given a set of features $X = \{\mathbf{x}^{(i)} \in \mathbb{R}^m\}_{i=1}^n$, task labels $Y = \{y^{(i)} \in \{1, \cdots, L\}\}_{i=1}^n$, and concepts $C = \{\mathbf{c}^{(i)} \in \{0,1\}^k\}_{i=1}^n$. Concepts are human-understandable attributes distilled from the raw input features $\mathbf{x}$. For example, if $\mathbf{x}$ is an image, then a concept could correspond to the attribute "Red Fruit". Each $\mathbf{c}^{(i)}$ is a 0-1 vector of length $k$, so that $c_j^{(i)} = 1$ indicates the presence of the $j$-th concept in $\mathbf{x}^{(i)}$.

We can use concept-based learning to construct interpretable architectures. A popular example of concept-based architectures is concept bottleneck models (CBMs) (Koh et al., 2020). CBMs are a two-stage architecture consisting of a concept predictor $g : \mathbf{x} \mapsto \hat{\mathbf{c}}$ and a subsequent label predictor $f : \hat{\mathbf{c}} \mapsto \hat{y}$. The concept predictor maps an input $\mathbf{x}$ to a set of interpretable concepts $\hat{\mathbf{c}}$, and then the label predictor uses these interpretable concepts to predict a label $\hat{y}$. During inference, for any input $\mathbf{x}$, a CBM predicts both a set of concepts $\hat{\mathbf{c}} = g(\mathbf{x})$ and a label $\hat{y} = f(g(\mathbf{x}))$.

The explainability of concept-based models arises from the concept predictions $\hat{\mathbf{c}}$. Typically, the label predictor $f$ is a shallow neural network (e.g., a linear layer), so mispredicted labels can be traced back

to concept prediction issues. For example, if an image is incorrectly predicted to be an "apple", then the incorrect prediction can be traced back to an incorrect prediction for the "Red Fruit" concept.

## 3.2 Locality and Concept-Based Explainability

To understand the assumptions underpinning concept-based models, we first introduce the idea of locality. We define *locality* as a dataset property where concepts can be fully predicted using some subset of features. For any concept and data point combination, we define its *local region* as a minimal set of features needed to fully predict the concept. Formally, for each concept $j$ and data point $\mathbf{x}^{(i)}$, we define its local region as $\mathcal{L}_j(\mathbf{x}^{(i)}) \subseteq \{1 \cdots m\}$, where $m$ is the number of features in $\mathbf{x}^{(i)}$. This corresponds to a minimum set of features that can completely predict the corresponding concept value. For example, the "Red Fruit" concept can be fully predicted using all the pixels from the foreground and none from the background. In Figure 1b, we highlight examples of local regions for different concepts.

In this work, we claim that concept-based models should make concept predictions based only on the features in a concept's local region. We believe that concept-based models can assist with interpretability, but only when the underlying concept predictors respect localities. When concept-based models fail to respect localities, they rely on features outside a concept's local region to make predictions. Because features outside of a concept's local region have no impact on the true presence of a concept, concept-based models may rely on spurious correlations to predict concepts. Concept prediction based on spurious correlations jeopardizes the use of concepts as an explainability mechanism because the presence of a concept can no longer reliably provide insight into a model's reasoning. For example, if the "Red Fruit" concept is predicted using irrelevant elements such as the presence of a "Fruit Bowl" rather than the presence of red fruit, then we cannot explain predictions based on the "Red Fruit" concept. A better understanding of localities allows for a better understanding of the trustworthiness of concept-based models.

We answer some common misconceptions about locality:

1. **Misconception #1: "Respecting localities and avoiding spurious correlations are the same problem"** - Spurious correlations and respecting localities are concerned with learning salient patterns. However, the motivations behind each are different; respecting localities is concerned with the explainability of concept-based models, while spurious correlations are concerned with the generalizability of models. Moreover, as we demonstrate in Section 5.1, even in datasets without shortcuts, and therefore, no motivations to learn spurious correlations, concept-based models can still fail to respect localities.

2. **Misconception #2: "Global concepts negate locality"** - If a concept is global and not confined to a local subset of features, then the local region is all the features; $\mathcal{L}_j(\mathbf{x}^{(i)}) = [m]$. In this scenario, all concept predictors respect localities because no features exist outside a concept's local region. In this work, we do not focus on such global concepts because they do not yield much insight into the nature of locality.

3. **Misconception #3: "Not respecting localities is a benefit, not a drawback of concept-based models"** - While leveraging correlations between features outside of the local region and the presence of a concept could lead to spurious yet accurate concept predictors, they endanger the separation between concepts and labels in the concept-based model. Combining label and concept predictions should reflect the presence of a concept rather than correlations. Otherwise, generating trustworthy explanations from concept predictions becomes infeasible and concept predictors become inherently less robust to changes to out-of-locality features.

4. **Misconception #4: "Concept leakage and locality refer to the same phenomenon"** - Concept leakage refers to the phenomenon where information about downstream labels or overall data distribution are "leaked" into concept predictions, which can reduce the purity of concept predictions (Mahinpei et al., 2021). While locality is also concerned with concept predictors, its focus is on the robustness of concept predictors to feature perturbations outside of a concept's local region. Locality assesses whether concept predictions truly reflect a model's reasoning about a concept, while leakage assess whether downstream labels leak into concept predictions.

# 4  Quantifying Locality

We assess the reliability of concept-based models by analyzing whether these models respect localities in a dataset. As noted in Section 3, concept-based models failing to respect localities jeopardizes the assumption that concept predictions are indicators of a model's reasoning behind a prediction. For example, if the prediction for the concept "Red Fruit" is not based on predicting red fruit but rather some correlated concept, then it becomes difficult to explain model predictions based on concept predictions. We study the extent to which models respect localities to understand whether we can trust the explanations and concept predictions from concept-based models.

We construct three metrics to quantify locality because each metric captures a notion of perturbation that concept-based models must be invariant to. When concept-based models respect localities, they are invariant to perturbations outside of a concept's local region. Each of our metrics for locality corresponds to a different notion of perturbation; locality leakage corresponds to adversarial perturbations, locality intervention corresponds to in-distribution perturbations, and locality masking corresponds to out-of-distribution masking perturbations. We present a pictorial example of each metric in Figure 1.

## 4.1  Locality Leakage

The locality leakage metric assesses a concept-based model's invariance to adversarial distortion outside of a concept's local region. Prior work has demonstrated that adversarial perturbations can impact a model's predictions (Goodfellow et al., 2014). However, it is unknown whether adversarial perturbations outside of a concept's local region impact their predictions, especially in scenarios without spurious correlations between features.

To formalize the impact of adversarial perturbations on a concept's predictions, we define the *locality leakage* metric as follows:

$$h_l\left(g, k, \{\mathbf{x}^{(i)}\}_{i=1}^n, \{\mathcal{L}_j\}_{j=1}^k\right) = \frac{1}{k}\sum_{j=1}^k \max_{\substack{\mathbf{x}'\in\mathbb{R}^m \text{ s.t.} \\ \forall a\in\mathcal{L}_j(\mathbf{x}^{(i)}),\ \mathbf{x}'_a=\mathbf{x}^{(i)}_a}} |g(\mathbf{x}')_j - g(\mathbf{x}^{(i)})_j| \qquad \text{(Locality Leakage)}$$

Here, $\mathbf{x}'$ is an adversarial data point, which perturbs features outside of the local region for concept $j$, $\mathcal{L}_j(\mathbf{x}^{(i)})$. For this data point, we measure the degree to which we can alter the prediction of concept $j$ (i.e., $|g(\mathbf{x}'_j) - g(\mathbf{x}^{(i)})_j|$) simply through adversarial perturbation of features outside of a concept's local region (i.e., $\forall a \in \mathcal{L}_j(\mathbf{x}^{(i)})$, $\mathbf{x}'_a = \mathbf{x}_a^{(i)}$). Taken together, the locality leakage metric averages across all concepts $j$, the maximum perturbation in concept prediction, $|g(\mathbf{x}'_j) - g(\mathbf{x}^{(i)})_j|$, by selecting an alternate data point $\mathbf{x}'$ that agrees on the local region $\mathcal{L}_j(\mathbf{x}^{(i)})$. That is, $\mathbf{x}'$ leverages features outside of its local region to modify the prediction of concept $j$. A large locality leakage implies that modifications to irrelevant features significantly impact a concept's prediction. In Appendix B.1, we consider an alternative formulation of the locality leakage metric which constrains the magnitude of adversarial perturbations and demonstrates similar results up to certain levels of constraints.

## 4.2  Locality Intervention

We introduce the locality intervention metric to assess a concept-based model's invariance to in-distribution shifts. The locality leakage metric potentially suffers from perturbations leading to significantly out-of-distribution features never seen in practice. To address this, we perturb features outside the local region while ensuring that the resulting feature vector $\mathbf{x}'$ is still in distribution and the underlying ground-truth concept is unchanged. To perturb a data point $\mathbf{x}^{(i)}$, we find an alternative data point $\mathbf{x}' = \mathbf{x}^{(l)} \in X$, so that concept $j$ is unaltered, $c_j^{(i)} = c_j^{(l)}$, while the prediction for concept $j$ is maximally altered, $|g(\mathbf{x}_j^{(l)}) - g(\mathbf{x}^{(i)})_j|$. Because $g(\mathbf{x}^{(l)})_j \in [0, 1]$ rather than being $\{0, 1\}$, the locality intervention metric allows us to quantify the magnitude of impact that out-of-local region perturbations in a more fine-grained manner compared to

concept prediction accuracy. Formally, we define the *locality intervention* metric as follows:

$$h_i\left(g, k, \{\mathbf{x}^{(i)}\}_{i=1}^n, \{\mathbf{c}^{(i)}\}_{i=1}^n\right) = \frac{1}{k} \sum_{j=1}^k \max_{\substack{\mathbf{x}^{(l)} \in X \text{ s.t.} \\ c_j^{(l)} = c_j^{(i)}}} |g(\mathbf{x}^{(l)})_j - g(\mathbf{x}^{(i)})_j| \qquad \text{(Locality Intervention)}$$

Here, we take the average perturbation $|g(\mathbf{x}^{(l)})_j - g(\mathbf{x}^{(i)})_j|$ across concepts $j$, by seeing how much an alternate data point $\mathbf{x}^{(l)}$ with the same ground-truth concept value disagrees in concept prediction. In essence, this measures a concept predictors susceptibility to in-distribution perturbations across other concepts outside of concept $j$'s local region.

### 4.3   Locality Masking

We introduce a third metric that assesses a concept-based model's invariance to a masking perturbation outside a concept's local region. The locality masking metric resides between the locality leakage and intervention metrics in terms of perturbation severity. The locality leakage metric adversarially perturbs non-local region features, which can lead to out-of-distribution data points, while the locality intervention metric ensures that perturbations are in distribution, but requires a large dataset $C$ to find in-distribution perturbations. To get around the limitations of locality leakage and intervention, we assess the impact masking concept $j'$ upon the prediction for concept $j$. By relying on masking to perturb features, our resulting features are not significantly out-of-distribution, especially because we only mask a fraction of the total features. All we require is access to the localities for each concept, through, for example, bounding boxers for concepts.

We assess two components for the locality masking metric: the *locality relevant masking metric* and the *locality irrelevant masking metric*. For a concept $j$, the locality relevant masking metric computes the impact of masking relevant features $\mathcal{L}_j(\mathbf{x}^{(i)})$, while the locality irrelevant masking metric computes the impact of masking features relevant to an unrelated concept $j'$, $\mathcal{L}_{j'}(\mathbf{x}^{(i)})$. Let $M(\mathbf{x}^{(i)}, j)$ be the replacement of all features in $\mathcal{L}_j(\mathbf{x}^{(i)})$ with a constant mask value $\eta$. Additionally, let $S(\mathbf{x}^{(i)}, j) = \{j' | \mathcal{L}_j(\mathbf{x}^{(i)}) \cap \mathcal{L}_{j'}(\mathbf{x}^{(i)}) = \emptyset\}$, which corresponds to the set of concepts such that the local region of $j'$ is non-overlapping with the local region for $j$. This way, masking concept $j'$ should not impact the prediction for concept $j$. Then, we define the locality relevant masking metric as:

$$h_{rm}\left(g, k, \{\mathbf{x}^{(i)}\}_{i=1}^n, M\right) = \frac{1}{k} \sum_{j=1}^k \left( |g(M(\mathbf{x}^{(i)}, j))_j - g(\mathbf{x}^{(i)})_j| \right) \qquad \text{(Relevant Masking)}$$

Similarly, we define the locality irrelevant masking metric as:

$$h_{im}\left(g, k, \{\mathbf{x}^{(i)}\}_{i=1}^n, M\right) = \frac{1}{k} \sum_{j=1}^k \frac{1}{|S(\mathbf{x}^{(i)}, j)|} \sum_{j' \in S(\mathbf{x}^{(i)}, j)} |g(M(\mathbf{x}^{(i)}, j'))_j - g(\mathbf{x}^{(i)})_j| \qquad \text{(Irrelevant Masking)}$$

Here, $M(\mathbf{x}^{(i)}, j)$ defines a masking function that masks concept $j$ in data point $\mathbf{x}^{(i)}$. We analyze the impact of masking upon the concept prediction, both when masking concept $j$ (where we compute $|g(M(\mathbf{x}^{(i)}, j))_j - g(\mathbf{x}^{(i)})_j|$), and when masking concept $j'$ (where we compute $|g(M(\mathbf{x}^{(i)}, j'))_j - g(\mathbf{x}^{(i)})_j|$). We note that we divide by $S(\mathbf{x}^{(i)}, j)$ to account for the number of masked concepts $j'$.

Concept-based models that respect localities should be unimpacted by masking outside of the local region and only impacted by perturbations to features within their local region. That is, such models should have a high locality relevant masking metric and a low locality irrelevant masking metric.

## 5   Experiments

We construct experiments to assess whether CBMs respect localities in both (a) controlled testbed where we can understand the impact of various factors and (b) real-world datasets where we can understand the performance of concept-based models in practice.

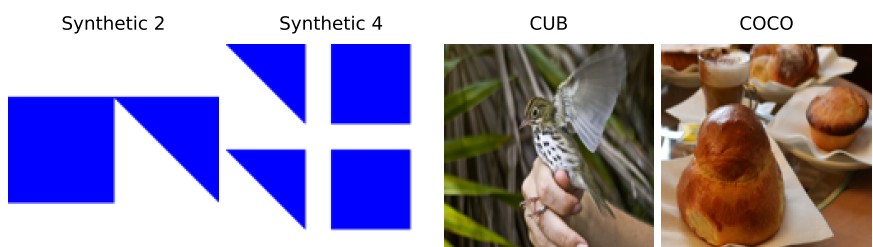

Figure 2: We plot examples from our 2-object and 4-object synthetic datasets, along with examples from CUB and COCO. For the synthetic datasets, concepts refer to the positions of each shape. For CUB, concepts are the colours and shapes of different parts of the bird, and for COCO, concepts are the presence of objects such as bowls and spoons.

### 5.1 Experimental Setup

**Synthetic Datasets**  We construct a synthetic dataset to understand locality in a controlled environment (see Figure 2). Our synthetic dataset consists of $k$ concepts and $\frac{k}{2}$ object locations. Each location consists of a triangle or a square, corresponding to two concepts: "Is Square" and "Is Triangle", for a total of $k$ concepts. For example, in the synthetic 2-object dataset, we have four concepts and two object locations. We construct datasets with 1, 2, 4, and 8 objects. For the 1- and 2-object datasets, we construct 256 training examples, while for the 4- and 8-object datasets we construct 1024 training examples due to the increased number of concept combinations. We construct our dataset using synthetic objects so concepts for different object locations are independent; the presence of a triangle on the left side gives no information on the shape of the right. This construction implicitly *should* discourage models from picking up spurious correlations due to the absence of shortcuts.

**Non-Synthetic Datasets**  To understand locality in more realistic scenarios, we evaluate concept-based models on two non-synthetic datasets: Caltech-UCSD Birds (CUB) (Wah et al., 2011) and Common Objects in Context (COCO) (Lin et al., 2014). We select these datasets because both are ubiquitous within concept-based explainability (Espinosa Zarlenga et al., 2022; Choi et al., 2023). The CUB dataset consists of images annotated with a bird species (the label) and properties, such as wing colour and beak length. This results in 112 attributes, selected by Koh et al. (2020), which can be partitioned into 28 concept groups based on the body part (e.g. wing). The COCO dataset consists of scenes with objects. The original COCO dataset consists of 100 objects, and we randomly downsample this to 10 objects so that COCO and CUB have comparable dataset sizes. Concepts refer to the presence of objects and the task label is a function constructed from this concept that indicates whether any object in some fixed subset is present. We construct a task in this manner following prior work (Espinosa Zarlenga et al., 2022).

**Other Experimental Details**  We vary our architecture and training procedure across experiments and detail these choices for each experiment individually. We run all experiments for three seeds on an NVIDIA TITAN Xp GPU, with the total number of hours ranging from 100 to 200. We run experiments on a Debian Linux platform with 4 CPUs and 1 GPU. We train our synthetic models for 50 epochs, our COCO model for 25 epochs, and our CUB model for 100 epochs, selecting each based on the time at which accuracy and loss reach convergence. For CUB and COCO we select a learning rate of 0.005, while for the synthetic experiments, we select 0.05, selecting this through manual inspection of model performance.

### 5.2 Impact of Architecture

We analyze how architecture choice impacts whether concept-based models respect locality. We perform our experiments across the different synthetic object datasets (see Section 5.2); we do so because we can quickly evaluate architectures without tuning parameters due to the simplicity of the dataset. Additionally, even smaller models can learn the synthetic dataset, while the same is not true for CUB and COCO. Here, we first evaluate models based on our locality leakage metric. To instantiate locality leakage, we employ projected

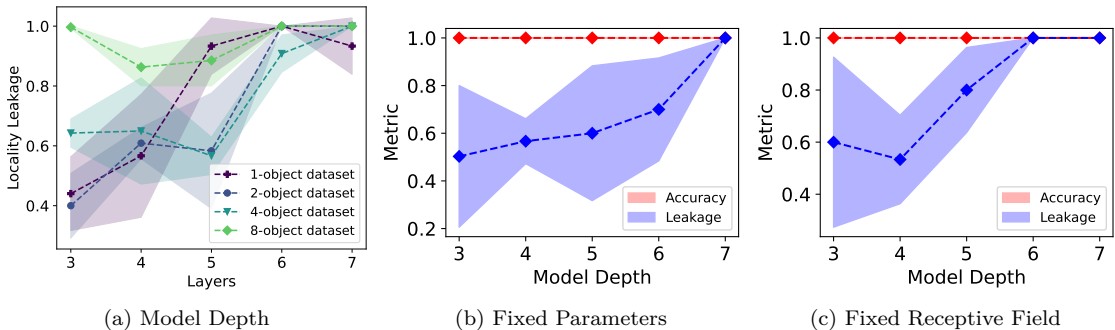

(a) Model Depth

(b) Fixed Parameters

(c) Fixed Receptive Field

Figure 3: We find that deep concept predictors lead to higher locality leakage, implying that deep concept predictors leverage features beyond a concept's local region. Such a trend occurs even when accounting for the number of parameters (b) or the receptive field size(c).

gradient descent to find the data point $\mathbf{x}'$ that maximizes $|g(\mathbf{x}')_j - g(\mathbf{x}^{(i)})_j|$ for each concept $j$. Projected gradient descent allows us to quickly find $\mathbf{x}'$, while ensuring that concepts within the local region $\mathcal{L}_j(\mathbf{x}^{(i)})$ remain constant. In Section 5.3, we consider our locality intervention and masking metrics.

Prior work in spurious correlations by Sagawa et al. (2020) demonstrates that larger models learn more spurious correlations, so we naturally investigate whether such a pattern holds for localities. We construct a set of concept predictors which vary in depth from 3 to 7 layers, where deeper models have more parameters. We train all models for 50 epochs; at 50 epochs, all models achieve perfect task and concept accuracy.

We compare the locality leakage for our models across varying model depths and find in Figure 3 that deeper models tend to have higher locality leakages. Across all models and dataset combinations, the locality leakage is at least 0.4, implying that on average, concept predictions can be changed by at least 40% through adversarial perturbations. We also find that increasing model depth leads to higher locality leakages, with 7-layer models having locality leakages over 0.9 for all synthetic datasets. This mimics the trend found in the spurious correlation literature where deeper models tend to produce more spurious correlations (Sagawa et al., 2020). Comparing across our synthetic datasets, we find that the 8-object dataset produces models with the highest leakage for all models except for 5-layer models. Our experiments reveal that deeper models, which also correspond to models with more parameters, tend to respect locality less.

While deeper models lead to higher levels of locality leakage, it is unclear whether this is due to (a) the concept predictors being deeper, (b) the concept predictors having more parameters, or (c) the concept predictors having a larger receptive field. To distinguish between these scenarios, we construct concept predictors that hold either (a) the number of parameters or (b) the receptive field size constant. We again vary the model depth between 3 and 7 layers for both settings and compare both the concept accuracy and the locality leakage metric for these models.

We find that depth, rather than the number of parameters or receptive field size after all convolutions, is primarily responsible for the increase in locality leakage. In Figure 3b and Figure 3c, we find that even when holding the number of parameters or receptive field size constant, depth-7 models still exhibit a locality leakage of 1.0. Between depth 4 and depth 7 we find that the locality leakage is non-decreasing, so increasing depth either increases locality leakage or keeps it at 1.0. Our results imply that overly deep models could be partially responsible for concept-based models failing to respect localities.

## 5.3 Dataset Choice

**Locality Intervention** To understand how concept-based models perform beyond synthetic datasets, we assess robustness to in-distribution (through locality intervention) and out-of-distribution (through locality masking) perturbations in real-world datasets. For in-distribution perturbations, we conjecture that training diversity plays a large role because more diverse training datasets allow concept predictors to better differentiate between concepts. To quantify this, we analyze the impact of train-time concept combinations

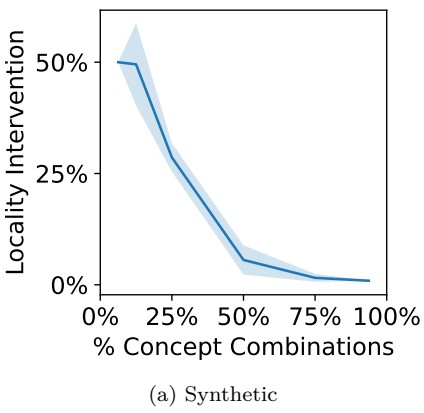

(a) Synthetic

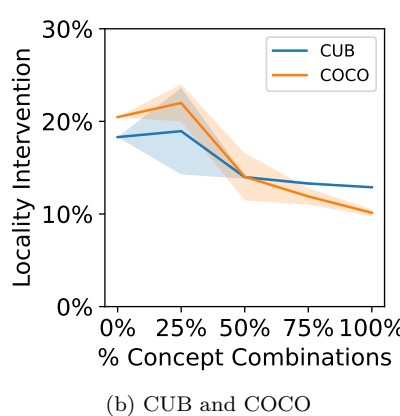

(b) CUB and COCO

Figure 4: We find that datasets with a larger number of concept combinations can reduce the locality intervention metric in synthetic (a) and real-world scenarios (b). Therefore, training with a diverse set of concept combinations can help improve the in-distribution robustness of concept predictors.

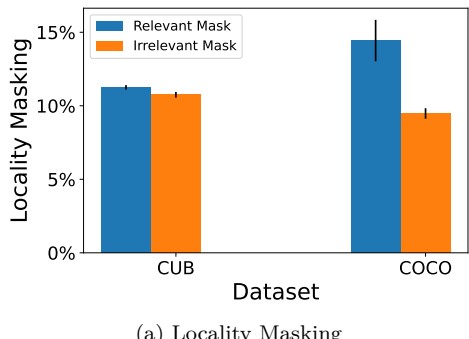

(a) Locality Masking

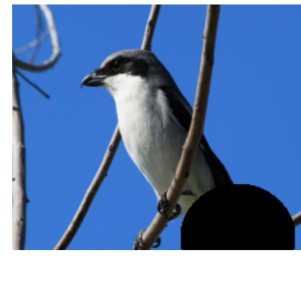

(b) Masking Example

Figure 5: To assess robustness to out-of-distribution perturbations, we compare the relevant and irrelevant masks for the CUB and COCO datasets. For CUB, masking relevant or irrelevant concepts similarly impacts predictions, indicating that concept predictors leverage features beyond a concept's local region to make predictions. For example, masking the tail of the bird (b) does not lead to changes in prediction for the concept of "Tail Colour."

on test-time locality intervention. We compare this trend for three datasets: the 8-object synthetic dataset, CUB, and COCO. For each dataset we vary the fraction of concept combinations from 25% to 100%. For example, 25% corresponds to sampling 25% of the values for $\mathbf{c}^{(i)}$ seen in the training dataset and filtering the training dataset to only use tuples $(\mathbf{x}^{(i)}, \mathbf{c}^{(i)}, y^{(i)})$ with corresponding $\mathbf{c}^{(i)}$. 100% corresponds to using the original training dataset. We ensure that perturbations do not impact concept labels by creating a set of data points for each concept combination $\mathbf{c}$. By doing so, we can always find a data point where concept $j$ is held constant, while other concepts $j'$ are varied.

We analyze the relationship between the diversity of the training concept and the locality intervention in Figure 4 and find that more combinations of concepts decrease the locality intervention. In the synthetic dataset, observing 50% of concept combinations reduces the locality intervention to 0.06. We see similar inflexion points for CUB and COCO as using 50% of concept combinations reduces locality intervention to 0.14. For all three data sets, increasing the diversity of the training set reduces locality intervention, demonstrating the importance of constructing data sets with a large number of combinations of concepts. Our results reflect that dataset-related changes can impact in-distribution perturbations, while architectural changes can impact out-of-distribution perturbations.

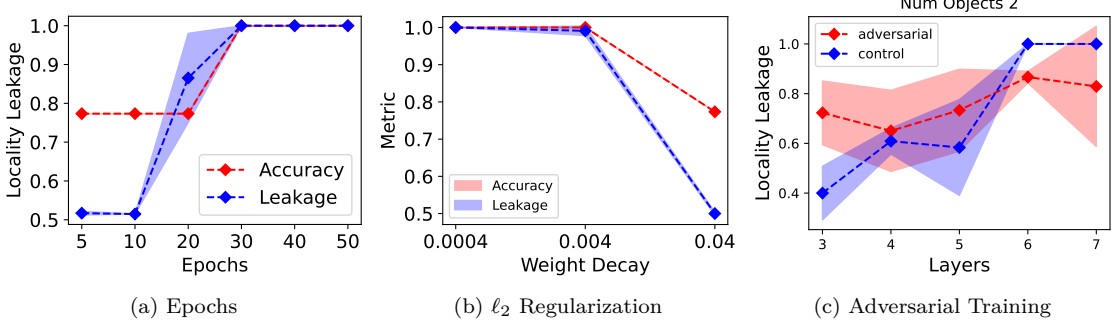

(a) Epochs  (b) $\ell_2$ Regularization  (c) Adversarial Training

Figure 6: We find that training time modifications such as (a) early stopping, (b) $\ell_2$ regularization, and (c) adversarial training cannot significantly reduce locality leakage without also impacting accuracy. Applying adversarial training leads to locality leakage values between 0.6 and 0.9 across all concept predictor depths, so all concept predictors exhibit some locality leakage.

**Locality Masking**  Next, we investigate the performance of concept-based models with real-world datasets through the locality masking metric. Locality masking captures more realistic out-of-distribution perturbations than locality leakage because the perturbations are not as dramatic, making it more appropriate for use with real-world datasets. For CUB, we have access to the centroid of each concept, so we mask all concepts within a radius of $\epsilon = 0.3l$, where $l$ is the image width (we evaluate the impact of varying $\epsilon$ in Appendix B.6). To avoid masking unrelated features, we mask only features closer to the centre of the masked concept than the centre of any other concept. For COCO we are provided the bounding box, so we use that instead. We select such an $\epsilon$ for CUB because it masks each concept without covering adjacent concepts; experiments with other values of $\epsilon$ reveal similar trends. We fill masked values with zero (black) but confirm our results using mean-masks in Appendix B.4.

In Figure 5, we assess locality masking for CUB and COCO. For CUB, we find only a 4.7% difference between relevant and irrelevant masks, indicating that concept predictors leverage features beyond a concept's local region to make predictions. In both COCO and CUB, the locality masking metric is at most 15%; intuitively, even when fully masking the concept, concept predictors can make predictions with only a 15% loss in accuracy. For example, masking the bird's tail in Figure 5b does not hinder tail colour prediction. This occurs because concept-based models infer a tail's colour from its body or head. Such behaviour is undesirable because concept predictors become detached from the concepts themselves, so predictions for the tail of a bird reflect properties of the body and head.

## 5.4 Training Modifications

We analyze the impact of modifications to training procedures, including the number of epochs and loss function, upon the locality leakage metric. Our results in Section 5.2 demonstrated that the selection of architecture impacted locality leakage, so it would be natural to understand whether training modifications also impact locality leakage. Prior work demonstrated that early stopping, which occurs when training stops after the loss converges or increases for a fixed number of epochs, can prevent overfitting (Caruana et al., 2000). We investigate whether such phenomena can impact locality, so we analyze the impact of training iterations upon locality. We vary the number of epochs from 5 to 50 and analyze the impact upon accuracy and leakage.

In Figure 6, we find that increasing the number of epochs leads to increased accuracy while also increasing locality leakage. When we train models for 30 or more epochs, they achieve perfect concept accuracy but also achieve a locality leakage of at least 0.9. Moreover, there exists no training setting where the model achieves perfect accuracy while having a small locality leakage.

We additionally investigate whether modifications to the training loss can improve the locality properties of concept-based models. We consider two types of training modifications: $\ell_2$ regularization (weight decay)

Table 1: Recent iterations of CBMs such as CEM and ProbCBM can reduce locality leakage for synthetic and real-world datasets due to their increased representational capacity. However, VLM-based approaches such as label-free CBMs fail to differentiate between relevant and irrelevant masking, showing that this model fails to address locality. We refrain from evaluating locality masking on the synthetic dataset because it would be redundant with the locality intervention metric.

| Dataset | CBM Variant | Accuracy | Leakage | Relevant Masking | Irrelevant Masking |
|---|---|---|---|---|---|
| Synthetic | CBM Baseline (7-Layer) | $100.0\% \pm 0.0\%$ | $1.000 \pm 0.000$ | $0.500 \pm 0.000$ | $0.052 \pm 0.037$ |
| | CEM | $100.0\% \pm 0.0\%$ | $0.724 \pm 0.028$ | $0.380 \pm 0.002$ | $0.355 \pm 0.012$ |
| | ProbCBM | $100.0\% \pm 0.0\%$ | $0.635 \pm 0.014$ | $0.384 \pm 0.000$ | $0.329 \pm 0.002$ |
| CUB | CBM Baseline (7-Layer) | $67.1\% \pm 0.4\%$ | - | $0.113 \pm 0.002$ | $0.107 \pm 0.002$ |
| | Label-Free CBM | $74.6\% \pm 0.0\%$ | - | $0.053 \pm 0.001$ | $0.052 \pm 0.001$ |
| | CEM | $68.1\% \pm 1.0\%$ | - | $0.035 \pm 0.001$ | $0.034 \pm 0.001$ |
| | ProbCBM | $68.2\% \pm 0.8\%$ | - | $0.008 \pm 0.001$ | $0.007 \pm 0.000$ |
| COCO | CBM Baseline (7-Layer) | $83.9\% \pm 0.3\%$ | - | $0.144 \pm 0.014$ | $0.095 \pm 0.004$ |
| | CEM | $83.6\% \pm 1.1\%$ | - | $0.024 \pm 0.009$ | $0.020 \pm 0.012$ |
| | ProbCBM | $83.3\% \pm 0.4\%$ | - | $0.024 \pm 0.003$ | $0.008 \pm 0.001$ |

and adversarial training. We select $\ell_2$ regularization because prior work showed that it can potentially reduce spurious correlations (Sagawa et al., 2019), while we select adversarial training because adversarially trained models are potentially more robust (Jin et al., 2020). For $\ell_2$ regularization, we vary the weight decay parameter in $\{0.0004, 0.004, 0.04\}$ and evaluate on the 2-object synthetic dataset with the 7-layer model. For adversarial training, we incorporate an adversarial loss and analyze the impact across different model depths in the 2-object synthetic dataset.

In Figure 6b and Figure 6c, we find that neither $\ell_2$ regularisation nor adversarial training can reduce locality leakage beyond 0.6 without impacting accuracy. While large amounts of $\ell_2$ regularization can reduce the locality leakage down to 0.5, we find that this also leads to a reduction in accuracy. When concept predictors have few layers, adversarial training leads to more locality leakage, while even for 6 and 7-layer concept predictors, the locality leakage is at least 0.8. This demonstrates that adversarial training fails to enforce that concept-based models respect localities.

## 5.5 CBM Variants

Newer CBM variants have alleviated issues ranging from representational power (Espinosa Zarlenga et al., 2022) to uncertainty (Kim et al., 2023), so we investigate the impact of these advances on our locality leakage metric. We evaluated the performance of three CBM variants according to accuracy and leakage (on the synthetic dataset) or masking (on the CUB and COCO datasets):

1. **Concept Embedding Models** address the low representational power of CBMs by allowing each concept to be represented through a vector (Espinosa Zarlenga et al., 2022).

2. **Probabilistic concept bottleneck models (ProbCBMs)** enable more accurate uncertainty estimates by modelling concept embeddings as normally distributed vectors (Kim et al., 2023).

3. **Label Free CBMs** incorporate large language models so CBMs can be trained without needing ground-truth labels for concepts. It does so by generating a concept set for a given list of labels and using the embeddings of this concept set to train label predictors (Oikarinen et al., 2023). We evaluate label-free CBMs only on the CUB dataset because they rely on human-understandable descriptions for each label, which are not present for the synthetic or COCO datasets.

We compare metrics across these CBM variants in Table 1. We find that CEMs and ProbCBMs reduce the leakage in the synthetic dataset, although a leakage of 0.6 still indicates that concept predictions can be modified by 60% of the original value. Moreover, ProbCBMs have a higher gap between relevant and irrelevant masking compared to the baseline CBM in the COCO dataset. Meanwhile, in the CUB dataset,

label-free CBMs only tighten the gap between relevant and irrelevant masking by a factor of 6. CEMs and ProbCBMs represent a potential panacea to address locality because they better model the embedding space for each concept, which assists concept predictors in differentiating between concepts. For example, ProbCBMs allow concept predictors to incorporate uncertainty when making concept predictions. With label-free CBMs, we find that the difference between relevant and irrelevant masking is almost nonexistent, implying that concept predictors leverage features beyond the local region. Our results with other variants of CBMs demonstrate that better-trained concept predictors, with better representational capacities, have the potential to reduce locality-related issues in some situations, though locality-related issues still persist with such models.

# 6 Theoretical Insights into Locality

To understand whether our empirical takeaways from Section 5 are an artefact of our experimental setup, we investigate the fundamental relationships between concept-based models and locality. We aim to formalize our study of locality intervention by analyzing how dataset construction impacts concept predictors.

Datasets with many concepts and few data points make it difficult to learn good concept predictors. For example, if a dataset only contains images of cherries and no images of strawberries, then the "Red Fruit" concept would be perfectly correlated with the "Circular Fruit" concept. This makes it difficult or impossible to distinguish these two concepts, which means that the underlying concept predictors do not correspond to the true concept. We formalize this idea through the theorem below:

**Theorem 6.1.** *Suppose out of $k$ total concepts, we can predict $m$, $\{\gamma_1, \gamma_2, \cdots \gamma_m\} \subset \{1, 2 \ldots k\}$, with perfect accuracy, $\mathbb{P}[\hat{\mathbf{c}}_j \neq \mathbf{c}_j] = 0 \forall j \in \{\gamma_1, \gamma_2, \ldots, \gamma_m\}$. Let $M_{j,q,i,r} = \mathbb{P}[\mathbf{c}_j = q | \mathbf{c}_i = r]$ be a correlation matrix, where $q, r \in \{0, 1\}$. For any concept whose value we do not know ($j \notin \{\gamma_1, \gamma_2 \ldots \gamma_m\}$), if for $s$ different triplets $(q, i, r)$, with $i \in \{\gamma_1, \gamma_2 \ldots \gamma_m\}$, we have (a) $M_{j,q,i,r} \geq 1 - \alpha$ and (b) $\mathbb{P}[\mathbf{c}_i = r] \geq \beta$, then there exists a concept predictor that achieves error $\epsilon = \mathbb{P}[\hat{\mathbf{c}}_j \neq \mathbf{c}_j] \leq \alpha + (1-\beta)^s$, using only information about the concepts $\{\gamma_1, \gamma_2 \ldots \gamma_m\}$, $g(\mathbf{x})_j = w(g(\mathbf{x})_{\gamma_1}, g(\mathbf{x})_{\gamma_2}, \ldots, g(\mathbf{x})_{\gamma_m})$ for the concept predictor $g$ and for some function $w$.*

We present some intuition and interpretation for our Theorem. In the theorem, $\gamma_1, \gamma_2, \ldots, \gamma_m \in [k]$ represent the $m$ concepts that are perfectly known and $M_{j,q,i,r} \in [0, 1]$ represents the probability that concept $j \in [k]$ has a value of $q \in [0, 1]$, given that concept $i \in [k]$ has a value of $r \in [0, 1]$. If concept $j$ is correlated sufficiently many other concepts $i$ then we can train a concept predictor with low error $\epsilon$. The error rate, $\epsilon$ is dictated by two factors: $\alpha$, which represents the correlation rate between concept $i$ and concept $j$ (that is, $\mathbb{P}[\mathbf{c}_j = q | \mathbf{c}_i = r] \geq 1 - \alpha$), and $\beta$, which represents the probability that concept $i$ is $r$ (that is $\mathbb{P}[\mathbf{c}_i = r] \geq \beta$). Our bound on $\epsilon$ is strict when $s$ is large, as in that scenario, $(1-\beta)^s$ is small, while the $\alpha$ term becomes tight. Our bound can additionally be modified even if concepts in $\{\gamma_1, \gamma_2, \ldots, \gamma_m\}$ are known up to error $\delta$. In that scenario, our bound then becomes $\epsilon \leq \alpha + \delta + (1-\beta)^s$, so we only pay a linear cost in the error rate.

Theorem 6.1 highlights the interaction between architecture, dataset, and locality. Intuitively, the theorem implies there exists concept predictors with low error that only leverage a subset of concepts for prediction. These concept predictors make predictions *irrespective of the true presence of a concept*. For example, this implies there exists low-error concept predictors for "Red Fruit" and "Circular Fruit" that only leverage the "Red Fruit" concept, so the concept predictor for "Circular Fruit" is made irrelevant of whether the image contains a circular fruit. We prove such a theorem by explicitly constructing the low error concept predictor and analyzing this predictor. Here, $\alpha$ corresponds to the level of independence between concepts, so low $\alpha$ corresponds to situations where concepts are highly correlated, and $\beta$ is the frequency at which concept $i$ is $r$. In essence, *datasets with highly correlated concepts can lead to concept-based models failing to respect localities*, as they fail to distinguish between concepts. Such results mirror the trends in Section 5, where increasingly diverse datasets can improve robustness to in-distribution perturbations.

# 7 Discussion

**Avoiding Locality in Practice** While vanilla CBM models generally fail to respect localities, we describe architectural and dataset choices that can help alleviate this issue. Both our empirical (Section 5) and

theoretical (Section 6) results demonstrate that larger and more diverse datasets can improve the adherence of concept-based models to localities found in the dataset. Concept-based models require datasets with different combinations of concepts so that concept predictors can distinguish between concepts. Moreover, our empirical results demonstrate that smaller models tend to exhibit less locality; this is partially because these models are less prone to overfitting the spurious correlations found in datasets. Finally, models with improved representational capacity can better distinguish between concepts, as shown through the improved performance of ProbCBM and CEM models. Taken together, we recommend the following: (a) practitioners ought to be cautious about interpreting concept predictions as the arbiter of truth, (b) when possible, the datasets used by practitioners should be large and contain diverse concept combinations so concept predictors can distinguish between concepts, and (c) practitioners should use smaller models and models with greater representational capacity (e.g., CEMs or ProbCBMs) when possible.

**Limitations and Future Work**   Our work is an initial inquiry into the relationship between locality and concept-based models which naturally leads to future work. We look at two real-world datasets for concept-based models; we select these datasets because of their ubiquity and use across concept-based learning tasks. Future work could build on this by considering more datasets including across other modalities, such as those for tabular data (Espinosa Zarlenga et al., 2023c). Additionally, future work could consider the application of our metrics to both the regression and generation settings. We can extend our metrics to the regression setting in a straightforward manner by modifying the label predictor to predict continuous values, while extension to the generative setting would require analogous metrics so modifications to concepts only modify relevant generated features. Our work considers a set of architectural and training-related modifications to assess their impact on locality. Our study is not exhaustive; there are always new training techniques and modifications to architectures that could potentially alleviate issues related to locality. However, we aimed to construct a thorough study that studied a variety of modifications to better understand locality and its relationship to the assumptions made by concept-based models.

## 8   Conclusion

Concept-based models rely on the assumption that concept predictions can give insight into a model's reasoning. We test whether this assumption holds by analyzing whether concept-based models respect locality. We construct three metrics to quantify locality and apply these across architectural and training configurations. We find that lack of respect for locality is a pervasive problem that is partially alleviated through diverse training data and smaller architectures. We complement our empirical results with a theoretical analysis that demonstrates how concept correlations can lead to concept-based models which fail to respect locality. Our study sheds light into the assumptions underlying concept-based models and gives insight into how concept-based models can be constructed with trustworthy explanations.

## Broader Impacts

Our paper analyses the robustness of concept predictors, which underpin the explainability of CBMs. Understanding and improving explainability allows for safer machine learning model deployment, and our work aims to improve such deployments through better-understood interpretability algorithms.

## Acknowledgements

The authors would like to thank Katie Collins, Andrei Margeloiu, Tennison Liu, and Xiangjian Jiang for their suggestions and discussions on the paper. Additionally, the authors would like to thank Zohreh Shams for their comments on the paper, and from the NeurIPS XAI Workshop reviewers for their insightful comments. During the time of this work, NR was supported by a Churchill Scholarship, and NR additionally acknowledges support from the NSF GRFP Fellowship. MEZ acknowledges support from the Gates Cambridge Trust via a Gates Cambridge Scholarship. JH thanks support from Cambridge Trust Scholarship.

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

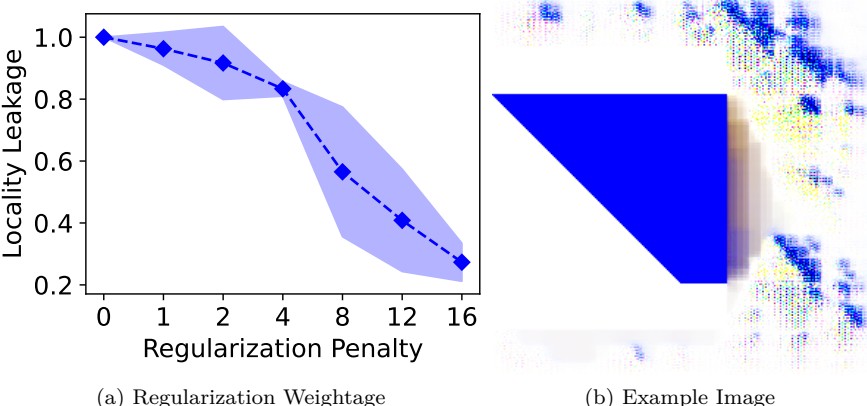

(a) Regularization Weightage        (b) Example Image

Figure 7: We introduce a regularization penalty to encourage perturbed images in locality leakage to remain similar to the original data point. While increasing the weight of this penalty decreases the locality leakage (a), we can find slightly perturbed examples that exhibit high locality leakage (b).

# A   Architecture Selection Details

In Section 5, we construct concept predictors of various depths and widths to analyze the impact of concept predictor architecture upon leakage. Our models increase in width from 64, for the 3-depth models, to 512, for the 7-depth models. For models which hold the number of parameters constant, we construct model layers so that shallower models are also wide; for example, the 3-layer model has widths 64, 64, and 32, while the 7-layer model has widths 64, 64, 64, 32, 28, 24, and 20. We similarly construct models to hold the receptive field size constant; details can be found in our code.

# B   Additional Ablation Results

## B.1   Constrained Locality Leakage

Because our locality leakage metric can lead to out-of-distribution inputs, we consider a variant which finds a perturbed datapoint $\mathbf{x}'$ while minimizing the distance from the original datapoint $\mathbf{x}$. We incorporate this as a penalty in the loss function, weighted by a hyperparameter $\lambda$. Here, $\lambda$ balances between significantly perturbed data points, and those with less perturbations but still achieving a high locality leakage (Figure 7a). As an example Figure 7b is a data point with less perturbation but with locality leakage near 1. More generally, we find that even when locality leakage outputs are highly constrained (large $\lambda$), there still exists perturbed data points that achieve high locality leakage.

## B.2   MLP Concept Predictors

To get more insight into the impact of concept predictor width and depth on locality leakage, we use multi-layer perceptrons (MLPs) as concept predictors. We vary the depth and width of MLPs, varying the depth from 1 to 3 and width from 5 to 15. In Table 2, we see that, when holding the depth constant at 1 or 3, increasing width leads to higher locality leakage. However, interestingly, holding the width constant and increasing depth generally leads to less locality leakage. Such a trend potentially occurs because of information bottlenecking (Tishby et al., 2000); increasingly deep models that are not sufficiently wide might make it difficult to train CBMs because information gets "stuck" between layers. The experiments with MLPs highlight that locality leakage can be an issue with other architectures, though the depth-width patterns from CNNs do not necessarily carry over to MLPs.

Table 2: With MLPs as concept predictors, wider MLPs lead to more locality leakage. However, deeper MLPs do not lead to more locality leakage, contradicting the pattern found in Section 5. This trend potentially occurs because deeper models lead to information bottlenecks which prevent information from travelling between layers (Tishby et al., 2000)

| Depth | Width | Locality Leakage |
|-------|-------|------------------|
| 1     | 5     | $0.63 \pm 0.32$  |
|       | 10    | $0.80 \pm 0.28$  |
|       | 15    | $\mathbf{0.97 \pm 0.05}$ |
| 2     | 5     | $0.59 \pm 0.30$  |
|       | 10    | $0.47 \pm 0.08$  |
|       | 15    | $0.57 \pm 0.32$  |
| 3     | 5     | $0.49 \pm 0.40$  |
|       | 10    | $0.74 \pm 0.19$  |
|       | 15    | $0.74 \pm 0.09$  |

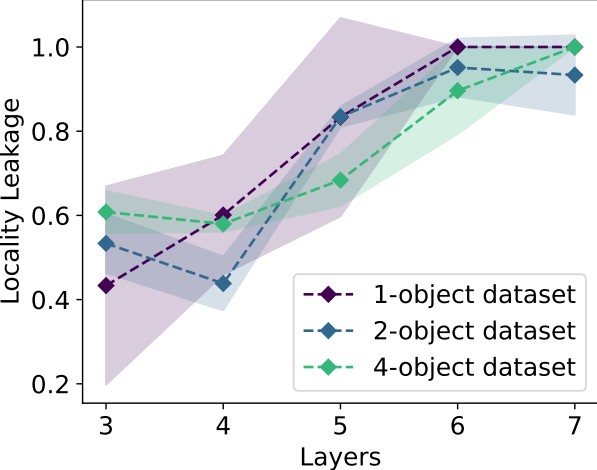

Figure 8: We reconstruct the experiment from Section 5 and adding noise to the synthetic datasets. We see that adding noise does not change the main result that deeper models lead to more locality leakage, which implies that our results are robust to noise in the input.

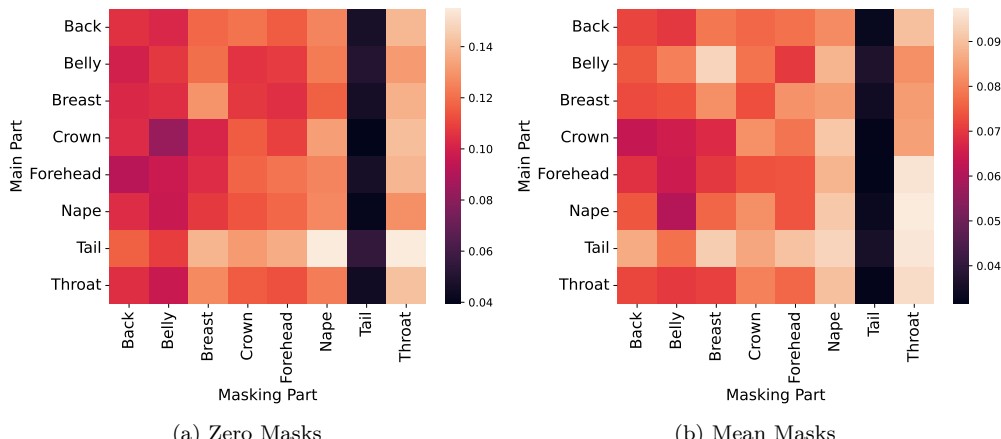

|  |  |
|---|---|
| (a) Zero Masks | (b) Mean Masks |

Figure 9: We compare the impact of zero masks (a) and mean masks (b) in the CUB dataset. We compute the change in prediction for one part (row) when masking another part (column). For both zero and mean masks, masking the belly has the same effect for predicting the belly as it does for predicting unrelated parts such as the back and forehead. Moreover, the absence of a strong diagonal pattern (where the same part is masked and analyzed) implies that both zero and mean masks have similar effects for relevant and irrelevant masking.

### B.3 Noisy Synthetic Datasets

We construct a variant of the synthetic dataset with additional noise to understand whether locality leakage persists when slightly perturbing data features. We construct such a dataset by adding Gaussian noise to the 1, 2, and 4-object datasets at train and test time, computing locality leakage across different layered concept predictors.

In Figure 8, we see that increasing the number of layers similarly leads to increased locality leakage. The trend matches the patterns seen in Section 5, where deeper concept predictors led to higher locality leakages. Similar results with or without noise imply that our results are not sensitive to small dataset perturbations.

### B.4 Selection of Masking Colour

We assess whether our locality masking experiments in Section 5 are a type of mask used. We originally used a zero mask, where features are modified to zero, and we assess whether similar results apply when using mean masks instead. The mean masks replace the data point features with the average value of the feature computed throughout the training set. By analyzing mean masks, we are able to see whether our choice of the masking colour has a big impact on our results.

We compare the results using zero and mean masks for the CUB dataset in Figure 9. We find that both zero and mean masks exhibit similar trends, where masking parts such as the back and belly can lead to performance decreases for unrelated parts. For example, masking the belly of a bird leads to a locality masking score of 0.10 for the back. Moreover, we see no difference between relevant and irrelevant masking, as shown by the absence of any pattern on the diagonal. Therefore, our locality masking results from Section 5 are not sensitive to the type of masks used, whether that is zero or mean masks.

### B.5 Residual Connections

Introducing residual connections in CBMs can alleviate concept leakage, so we explore whether similar choices can improve locality (Havasi et al., 2022). Residual connections introduce a connection between the input **x** and the output $y$, avoiding the bottleneck layer **c**. We introduce a similar connection and assess the impact on locality leakage in CBMs in a synthetic dataset.

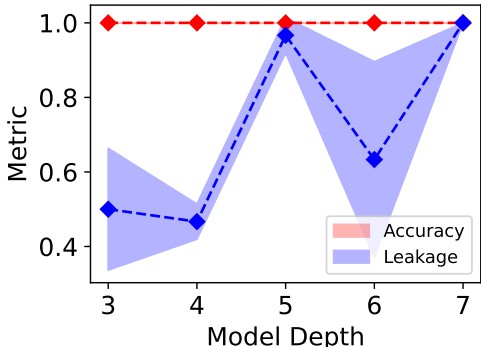

Figure 10: We incorporate residual connections following prior work Havasi et al. (2022) to understand whether techniques used to address concept leakage can also improve locality. Across model sizes, we find that models trained with residual connections still exhibit high levels of locality leakage, with this best seen with depth-7 models.

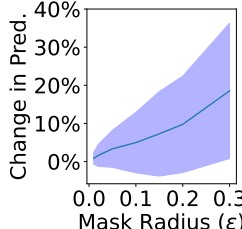

Figure 11: We vary the mask radius for the *tail* concept in the CUB dataset and find that, as expected, increasing mask radius leads to increasing change in concept prediction. However, even large masks that lead to significantly distorted images, hinting that predictors may use correlations to predict *tail colour*.

In Figure 10, we find that residual connections fail to impact locality leakage. Concept predictors with depths 5 and 7 still exhibit a locality leakage near 1 when using residual connections. The result demonstrates the difference between locality and concept leakage; solutions to concept leakage such as residual connections fail to improve the locality properties of CBMs.

### B.6  Impact of Mask Size

To better understand locality masking, we analyse the impact of mask size $\epsilon$ on concept predictions. We evaluate the impact of mask predictions in CUB for the concepts of *tail colour* and *tail shape*, selecting these since tails are visually distinct from other parts in this task. We reformulate masking in CUB so all features within $\epsilon$ of the part's centre are masked, even if the features are closer to another concept, and vary $\epsilon$ from 1% to 30% of $l$. Intuitively, we expect that increasing $\epsilon$ should increase changes in concept prediction, yet it is unclear how large $\epsilon$ needs to be so that models are significantly less confident. We compute the change in concept prediction across concept values that are at least 75% confident, selecting this as a threshold indicative of a high-confidence prediction, with the expectation that masking should decrease model confidence so models are uncertain (around 50%).

In Figure 11, we find that, as expected, increasing $\epsilon$ increases changed predictions, yet even large values of $\epsilon$ fail to change high-confidence concept predictions significantly. This implies that models are exploiting correlations between, for example, "*chest colour*" and "*tail colour*" to predict the "*white tail colour*" concept. Such a phenomenon might occur because concepts are occluded for some samples in CUB. For example, predicting "*tail colour*" when the tail is occluded requires leveraging "*chest colour*".

## B.7 Comparison with Other Metrics

We present a discussion of the relationship between our work and a set of related but distinct metrics proposed by Espinosa Zarlenga et al. (2023a). In that work, the authors propose two metrics: the oracle impurity score (OIS) and niche impurity score (NIS). OIS captures the ability of concept predictions for concept $j$ to predict an unrelated concept $j'$. Higher values for OIS indicate that more information is leaked between concepts, or that concept predictors are encoding impurities. NIS extends this to consider *sets* of concepts that encode impurities by investigating the predictive power of such sets. NIS values near $1/2$ indicate that no impurities are encoded, while larger NIS values imply that higher levels of impurity are encoded. For further details and a rigorous definition, we refer readers to prior work (Espinosa Zarlenga et al., 2023a).

We demonstrate that the OIS and NIS metrics fail to distinguish between models with low and high levels of locality leakage. We do so by comparing the OIS and NIS metrics across models of varying depth trained on the 2-object dataset. Across all model depths, we find that the OIS values range between 0.05 and 0.10, while the NIS values range between 0.41 and 0.47. Both metrics indicate that models do not leak impurities into concept predictors. Despite this, we find a large variation in locality leakage between models with 3 layers and those with 7. The reason for this discrepancy is because the OIS and NIS metrics are computed by comparing concept predictions to groundtruth concepts, which ignores whether concept predictors leverage the appropriate set of features in a concept's local region. In contrast, our metrics evaluate concept predictors across perturbed data points to understand whether these concept predictors respect locality. Our locality metrics are complementary to the OIS and NIS metrics, and they each measure different desirable properties of concept predictors.

## C   Proofs

This section presents our proof for Theorem 6.1 discussed in Section 6. For us to present our proof, we begin by showing two lemmas that will become useful later on. The first lemma, shown below, introduces a simple identity on sets of real numbers in $[0, 1]$:

**Lemma C.1.** *Consider a set of $n$ real numbers $\{p_1 \cdots p_n\}$, where $\forall i, \ 0 \leq p_i \leq 1$. Then, the following identity must hold:*

$$\sum_{i=1}^{n} p_i \prod_{j=1}^{i-1} (1 - p_j) = 1 - \prod_{i=1}^{n} (1 - p_i)$$

*Proof.* To simplify this proof, we will show the equivalent but more accessible statement that:

$$\prod_{i=1}^{n} (1 - p_i) = 1 - \sum_{i=1}^{n} p_i \prod_{j=1}^{i-1} (1 - p_j)$$

We proceed to do this by induction on the number of real numbers $n$. Our base case $n = 1$ trivially follows by noting that both sides of the equation above resolve to $1 - p_1$ when $n = 1$.

Now, assume that the claim above holds for all positive integers $n$ up to some value $k \in \mathbb{Z}^+$. Consider what happens when $n = k + 1$:

$$\prod_{i=1}^{k+1}(1-p_i) = (1 - p_{k+1})\prod_{i=1}^{k}(1 - p_i)$$

$$= \prod_{i=1}^{k}(1 - p_i) - p_{k+1}\prod_{i=1}^{k}(1 - p_i)$$

$$= \Big(1 - \sum_{i=1}^{k} p_i \prod_{j=1}^{i-1}(1 - p_j)\Big) - p_{k+1}\prod_{i=1}^{k}(1 - p_i)$$

$$= 1 - \Big(\sum_{i=1}^{k} p_i \prod_{j=1}^{i-1}(1 - p_j) + p_{k+1}\prod_{j=1}^{k}(1 - p_j)\Big)$$

$$= 1 - \sum_{i=1}^{k+1} p_i \prod_{j=1}^{i}(1 - p_j)$$

where the inductive hypothesis was applied in our second step. This shows that our statement holds for $n = k + 1$ and therefore holds for all positive integers $n$. $\qquad\square$

The second lemma directly builds on the identity introduced in Lemma C.1 to produce a simple upper bound for a sum of products involving the same list of reals in $[0, 1]$:

**Lemma C.2.** *Consider a set of $n$ real numbers $\{p_1 \cdots p_n\}$, where $\forall i$, $0 \le p_i \le 1$. Then*

$$\sum_{i=1}^{n} p_i \prod_{j=1}^{i-1}(1 - p_j) \le 1$$

*Proof.* We begin by noticing that, because all $\{p_1, \cdots, p_n\}$ are reals such that $\forall i$, $0 \le p_i \le 1$, each of the terms in the product $\prod_{i=1}^{n}(1-p_i)$ must be non-negative. This trivially implies that the product $\prod_{i=1}^{n}(1-p_i)$ itself must be non-negative. Therefore, we can conclude that:

$$0 \le \prod_{i=1}^{n}(1 - p_i)$$

$$\Leftrightarrow 1 - \prod_{i=1}^{n}(1 - p_i) \le 1$$

$$\Leftrightarrow \sum_{i=1}^{n} p_i \prod_{j=1}^{i-1}(1 - p_j) \le 1 \qquad\qquad \text{(by Lemma C.1)}$$

which is exactly what we wanted to show. $\qquad\square$

Equipped with these two lemmas, we now present a proof for the key theoretical result introduced in Section 6:

**Theorem 6.1.** *Suppose out of $k$ total concepts, we know $m$, $\{\gamma_1, \gamma_2, \cdots \gamma_m\} \subset \{1, 2 \cdots k\}$, with perfect accuracy. Let $M_{j,q,i,r} = \mathbb{P}[\mathbf{c}_j = q | \mathbf{c}_i = r]$ be a correlation matrix, where $q, r \in \{0, 1\}$. For any concept $j \notin \{\gamma_1, \gamma_2 \cdots \gamma_m\}$ whose value we do not know, if for $s$ different triplets $(q, i, r)$, with $i \in \{\gamma_1, \gamma_2 \cdots \gamma_m\}$, we have i) $M_{j,q,i,r} \ge 1 - \alpha$ and ii) $\mathbb{P}[\mathbf{c}_i = r] \ge \beta$, then there exists a concept predictor that achieves error $\epsilon = \mathbb{P}[\hat{\mathbf{c}}_j \ne \mathbf{c}_j] \le \alpha + (1 - \beta)^s$ using only information about the concepts $\{\gamma_1, \gamma_2 \cdots \gamma_m\}$.*

*Proof.* We propose an algorithm which achieves error rate $\epsilon = \mathbb{P}[\mathbf{c}_j = r] \le \alpha + (1 - \beta)^s$. To predict $\mathbf{c}_j$, we leverage the correlation between $\mathbf{c}_j$ and an $s$ different values of $M_{j,q,i,r}$, where $i \in \{\gamma_1, \gamma_2 \cdots \gamma_m\}$. By leveraging this, we're able to accurately predict the presence of $\mathbf{c}_j$

For this, let the $s$ triplets satisfying (i) and (ii) be $\mathcal{B} := \{B_l \mid B_l = (q_l, i_l, r_l)\}_{l=1}^s$. Furthermore, for any triplet $B_l = (q_l, i_l, r_l)$ in this set, let $p_l$ be $\mathbb{P}[\mathbf{c}_{i_l} = r_l]$.

Here, we propose an algorithm that takes as an input the set of triplets $\mathcal{B}$, along with their associated probabilities $\{p_l\}_{l=1}^s$, and returns a label $\hat{\mathbf{c}}_j$ for an unknown concept $j$. Our algorithm proceeds as follows: We will iterate with an index variable $u$ starting with $u = 1$ and finishing with $u = s$. At each iteration, we select the triplet $B_u = (q_u, i_u, r_u)$. As stated above, we know $i_u \in \{\gamma_1, \gamma_2 \cdots \gamma_m\}$, and therefore we have perfect knowledge of $\mathbf{c}_{i_u}$ Therefore, we can check if $\mathbf{c}_{i_u} = r_u$. If that is the case, then our algorithm predicts $\hat{\mathbf{c}}_j = q_u$. Otherwise, we continue to the next iteration of the loop. If we reach the end of the loop, then we simply guess that $\hat{\mathbf{c}}_j = 0$.

We note that our algorithm will terminate at step $u$ with probability $p_u \prod_{l=1}^{u-1}(1 - p_l)$. This is because to terminate at step $u$, we must encounter the event $\mathbf{c}_{i_u} = r_u$, which occurs with probability $\mathbb{P}[\mathbf{c}_{i_u} = r_u] = p_u$. Additionally, we must not terminate in any previous steps $l < u$, each event happening with probability $\mathbb{P}[\mathbf{c}_{i_l} \neq r_l] = 1 - \mathbb{P}[\mathbf{c}_{i_l} = r_l] = 1 - p_l$. If we terminate at step $u$, then we know that $\mathbf{c}_{i_u} = r_u$ and we predict $\hat{\mathbf{c}}_j = q_u$. This prediction fails with probability $\mathbb{P}[\mathbf{c}_j \neq q_u | \mathbf{c}_{i_u} = r_u] = 1 - \mathbb{P}[\mathbf{c}_j = q_u | \mathbf{c}_{i_u} = r_u] = 1 - M_{j,q_u,i_u,r_u}$. For the sake of simplifying notation, and because $j$ remains constant and implicitly known throughout the algorithm, we use $M_{B_u}$ to represent $M_{j,q_u,i_u,r_u}$ for triplet $B_u = (q_u, i_u, r_u)$.

The error rate $\epsilon = \mathbb{P}[\hat{\mathbf{c}}_j \neq \mathbf{c}_j]$ of this algorithm is then given by

$$\epsilon = \sum_{u=1}^s (1 - M_{B_u}) p_u \prod_{l=1}^{u-1}(1 - p_l) + \mathbb{P}[\mathbf{c}_j \neq 0] \prod_{u=1}^s (1 - p_u)$$

$$\leq \sum_{u=1}^s (1 - M_{B_u}) p_u \prod_{l=1}^{u-1}(1 - p_l) + \prod_{u=1}^s (1 - p_u)$$

as $\mathbb{P}[\mathbf{c}_j \neq 0] \leq 1$ Then, we note that $p_u \geq \beta$, implying that $\prod_{u=1}^s (1 - p_u) \leq (1 - \beta)^s$. Finally, we note that $1 - M_{B_u} \leq \alpha$, so $\sum_{u=1}^s M_{B_u} p_u \prod_{l=1}^{u-1}(1 - p_l) \leq \alpha \sum_{u=1}^s p_u \prod_{l=1}^{u-1}(1 - p_l) \leq \alpha$ by Lemma C.2. Combining all of the above gives us an error $\epsilon \leq \alpha + (1 - \beta)^s$.

$\square$

