# OpenReview forum: "Do Concept Bottleneck Models Respect Localities?"
_TMLR — Accepted by TMLR_

### Review · Reviewer_6Tsj · 2025-03-24

**Summary Of Contributions:**

This paper designs three metrics to quantify locality and evaluates them across different architectural and training configurations. The results indicate that the neglect of locality is a widespread issue, but this can be partially mitigated through the use of diverse training data and smaller model architectures. Additionally, the paper provides a theoretical analysis that reveals how concept correlations can lead to concept-based models failing to meet locality requirements.

**Audience:**

Yes

**Claims And Evidence:**

Yes

**Requested Changes:**

1. It is recommended that the authors reorganize the abstract and introduction, adding more details about the methodology to highlight its features and innovations.

2. It is suggested that the authors include more comparative methods, evaluation metrics, and dataset validation.

3. The article involves a large number of formulas and symbols, with particularly confusing use of superscripts and subscripts. It is recommended that the authors add a notation section or provide a table to clarify these elements. For instance,  Mj,q,i,r......

4. The description of the methodology in the article is not detailed enough. It is recommended to add a framework diagram of the method and pseudocode for the core steps.

**Strengths And Weaknesses:**

Strengths:
1. The main body of this paper is well-developed, with standardized figures and tables. Theoretical validation has been conducted for the core contributions.
2. The motivation and innovation of this paper are acceptable.

Weaknesses:
1. The writing of the abstract is extremely difficult to comprehend. It extensively discusses the research background but only briefly mentions the methodology in a single sentence. There is no mention of methodological innovation or how the experiments validate the findings. The issues raised in the background section are also quite peculiar. The author argues that conceptual-level explanations are inferior because they are "virtual," but isn't this precisely the advantage of conceptual-level explanations? They transform abstract features into concepts that are easier for humans to understand.

2. The author's writing and description are inadequate, leading to many logical doubts. The formulas are not numbered, and many symbols are not explained. Additionally, many formulas do not seem to be used later in the text. It is recommended that the author further organizes and clarifies these aspects.

3. The derivation of formulas in the article is relatively brief, lacking detailed intermediate steps. To enhance readability, the authors should provide more detailed derivation steps, especially for the intermediate processes of key formulas. The assumptions of Theorem 6.1 are quite stringent, which may be difficult to satisfy in practical applications, as real-world data often contains noise and incompleteness. Additionally, the specific form of the data distribution is not discussed in the assumptions of the theorem. The application of metrics such as "locality leakage" and "locality intervention," defined in the formulas, is not thoroughly explained in the experimental section.

4. This paper has shortcomings in the comparative validation of interpretability methods, which mainly include two aspects: 1）The paper should compare with more interpretability methods and adopt a broader range of evaluation metrics. 2）More experiments should be designed to verify the effectiveness of the proposed method. The current experimental setup is insufficient to fully validate the claims. It is recommended that the authors refer to other interpretability method papers to enrich the experiments. The paper primarily utilizes synthetic datasets and two real-world datasets for experiments. Although these datasets are widely used in the study of CBM, they do not comprehensively represent all possible application scenarios. The experiments in the paper mainly focus on variations in model architecture and training methods, lacking in-depth exploration of other factors that may influence model interpretability. Additionally, the experiments do not consider the impact of different task types (such as classification, regression, and generative tasks) on model interpretability. The absence of these factors limits the applicability of the experimental results, making it difficult to fully reflect the model's performance in real-world applications.

---

> ### Author Response · Authors · 2025-05-05
> **Author Response #1**
>
> Dear Reviewer 6Tsj,
>
> We thank you for your very in-depth suggestions and comments. Additionally, we appreciate that you find our work “well-developed” and its motivation and innovation appropriate for TMLR. We have updated our manuscript based on your suggestions and have marked any changes made in red. We provide answers to your questions below:
>
> ## Abstract Details
>
> We thank the reviewer for their suggestions on the abstract. We have updated the abstract with more details on our proposed metrics. We stress that our main contribution of this work is the creation of a new set of metrics and analysis of locality in concept-based models, together with pointing out a critical flaw in concept-based models and theoretically studying this limitation. We back this up through extensive experimentation to understand how locality varies across different dataset/model combinations.
>
> ## Importance of Concept-Level Explanations
>
> We agree with the reviewer that concept-level explanations are critical and make it easier for humans to understand the internals of machine learning models. We clarify that we do not claim that concept-based explanations are inferior. Rather, our concern is with the faithfulness of concept predictors and whether these concept predictors reflect the presence of the underlying. Our claim is that to ensure the explainability of concept-based models, we need concept predictors to respect localities or make predictions using the appropriate set of features. To take this feedback into consideration in our manuscript, we add extra details to Section 3.2 emphasizing that our goal is to better understand concept-based models and their explainability.
>
> ## Derivation and Description of Metrics
>
> We introduce four equations in the body of our paper, where each equation corresponds to one metric. Each of these equations is labelled by the corresponding metric, and these same four metrics are used in practice in Section 5. Essentially, each metric considers the maximum distortion in concept prediction according to different definitions of perturbation. In particular, the locality leakage metric adversarially perturbs the data point, the locality intervention metric perturbs the other concepts for a counterfactual intervention, and the locality masking metric applies a mask. For each of these metrics, we provide a more in-depth discussion of the derivation of these metrics in an updated version of Section 4. In response to your suggestion, we note that we have a comprehensive framework diagram in Figure 1 that illustrates the relationships between our three locality metrics. We modify our text to refer back to Figure 1 in both the introduction, and the description of the metrics in Section 4, and in our experiment in Section 5.
>
> ## Experimental Description of Metrics
>
> We thank the reviewer for this suggestion. We have updated Section 5.2 and Section 5.3 with additional details on how we implement locality leakage and locality intervention. To construct adversarial perturbations for our locality leakage metric, we optimize for an input $\\mathbf{x}^\\prime$ that does not modify features in its local region $\\mathcal{L}(\\mathbf{x}^\\prime)$, while maximally modifying the concept prediction $g(\\mathbf{x}^\\prime)\_{j}$. We do this through projected gradient descent, using such a method to optimize over feasible $\\mathbf{x}^\\prime$ given a sample $\\mathbf{x}$ [1]. We have included this information in an updated version of Section 5.2.
>
> We ensure that the perturbations for locality intervention do not impact the true concept label by constructing a dataset with many different combinations of concepts. We can do so in a synthetic controlled environment to simulate different combinations of concepts accordingly. We can then precompute the impact of perturbing a concept by storing data points for different concept combinations, as this ensures that perturbations do not impact concept labels. More generally, we ensure that we do not change concepts during perturbation because we rely on alternate data points found in our dataset with the specified concept value, which ensures that the concept label remains unchanged. We include these details in Section 5.3 of our updated manuscript.
>
> [1] Madry, Aleksander, et al. "Towards Deep Learning Models Resistant to Adversarial Attacks." International Conference on Learning Representations. 2018.

---

> ### Author Response · Authors · 2025-05-05
> **Author Response #2**
>
> ## Realistic Bound and Extension to Imperfect Concepts
>
> We thank the reviewer for asking about an extension to the imperfect concept predictor scenario. In this scenario, we identify three sources of errors:
> With probability $(1-(1-\\beta)^{s}) \\delta$, the concept $i$ is incorrectly predicted
> With probability $(1-(1-\\beta)^{s}) (1-\\delta)\\alpha$, $\\mathbb{P}[\\mathbf{c}\_{j}=q | \\mathbf{c}\_{i}=r]$ is incorrect (that is $\\mathbf{c}\_{i}=r$, but $\\mathbf{c}\_{j} \\neq q$)
> With probability (1-\\beta)^{s}, no such triplet is found.
>
> We can then bound $\epsilon$ as $(1-(1-\\beta)^{s}) (1-\\delta) \\alpha + (1-(1-\\beta)^{s})\\delta + ½*(1-\\beta)^{s}$, which we can further upper bound as $\\alpha + \\delta + (1-\\beta)^{s}$. Essentially, with errors of size $\\delta$, we loosen our bound by $\\delta$. We have incorporated this additional extension into Section 6 of our manuscript.
>
> Our theorem is developed with the intention that there's no assumption on the data distribution, just the correlations between concepts. By avoiding assumptions on the specific data distribution, we can give intuition on how models can learn unreliable concept predictors even in theory.
>
> ## Comparison with Existing Metrics
>
> Our primary contribution in this work is a new set of metrics and an analysis of locality using these metrics. While we cannot use other metrics to assess our metric (because our contribution in the paper is a set of new metrics), we do assess this new metric across a variety of concept-based models (e.g., ProbCBMs and label-free CBMs).
>
> We thank the reviewer for bringing up the suggestion to compare with other metrics. We will first contrast our work with [1], which proposes oracle and niche impurity scores (OIS and NIS, respectively) to measure the level of impurity or leakage contained within concept predictions. While both our metric and theirs are concerned with analyzing the effectiveness of concept-based models, our metric is more concerned with the features leveraged by concept predictors, while [1] is more concerned about leaking information into concept predictions. Moreover, in Appendix B.7, we include a comparison with the OIS and NIS metrics, where we demonstrate that the OIS and NIS metrics cannot distinguish between predictors that exhibit differing levels of locality leakage. In essence, the metrics proposed in [1] complement our metrics and measure different desiderata for concept predictors.
>
> We also contrast our work with [2], which focuses on shortcut learning in Neuro-symbolic settings. Both our work and [2] are focused on understanding shortcuts in interpretable models. However, we differ in the setting; we focus on the concept-based learning setting, while [2] focuses on Neuro-symbolic models. Moreover, [2] focuses on shortcut learning. In contrast, we focus on understanding concept-feature relationships, which might cause concept predictors to learn shortcuts, but not necessarily always (e.g., there are no shortcuts to learn in the synthetic dataset). We include a discussion of these two related works in an updated Section 2.
>
> Besides these external metrics, other notions of concept faithfulness include standard metrics such as concept accuracy or mean concept AUC-ROC. These metrics, however, are interested in measuring the faithfulness of concept predictions rather than whether they are faithfully capturing the appropriate concept (meaning they could not discriminate between a model exploiting spurious correlations to predict a concept and one that does not exploit those spurious correlations).
>
> [1] Zarlenga, Mateo Espinosa, et al. "Towards robust metrics for concept representation evaluation." Proceedings of the AAAI Conference on Artificial Intelligence. Vol. 37. No. 10. 2023.
>
>
> [2] Marconato, Emanuele, et al. "Not all neuro-symbolic concepts are created equal: Analysis and mitigation of reasoning shortcuts." Advances in Neural Information Processing Systems 36 (2023): 72507-72539.

---

> ### Author Response · Authors · 2025-05-05
> **Author Response #3**
>
> ## Effectiveness of our Method
>
> Our paper’s goal is to identify a limitation of concept-based models and develop a set of metrics to better understand this limitation. By doing so, we can understand whether concept predictors are faithful to the underlying concept that they represent. As such, our primary contribution is the analysis of existing methods and a new metric, rather than the creation of a new method.
>
> We use this metric to evaluate a set of common CBM models across standard CBM datasets. We focus on classification tasks because those are the most common in the CBM literature; however, even within classification, we consider both binary and multiclass classification. Moreover, our proposed metrics and analysis can naturally be applied to regression or generative concept-based models. We evaluate our metrics across the CUB and COCO datasets because they are popular CBM datasets on which models are frequently developed. On all datasets, we give a comprehensive overview of when concept predictors respect localities, and we complement this with a theoretical characterization of this idea. Finally, our theoretical contributions in Section 6 help complement our empirical studies to better understand locality in concept-based models. To reiterate, our aim with this work was not to propose a new method, but rather to investigate and understand the phenomenon of locality across a variety of settings in concept-based models, in both synthetic and real-world datasets.
>
> ## Variations beyond Architecture
>
> In our Appendices, we assess the locality of concept-based models in other settings beyond changes in architecture. These include variations around noisy data and constrained locality leakage, or when the magnitude of perturbations is limited (in Appendix B.1). We try out these varieties of architecture, dataset, and training dynamics to capture as many realistic use cases as we can, especially those commonly used in the concept-based model literature.
>
> While our current experiments focus on classification tasks (both binary and multi-class), as these dominate the CBM literature [1,2,3,4], we acknowledge the reviewer's point about task diversity. In our revised manuscript, we have added a discussion in Section 7 about how our metrics can be extended to regression and generative tasks. Specifically, for regression tasks, our locality metrics can be adapted by modifying the label predictor to use continuous predicted values. For future work, we outline a path to applying our framework to these additional task types, which we believe is a promising direction but beyond the scope of this paper.
>
> [1] Koh, Pang Wei, et al. "Concept bottleneck models." International conference on machine learning. PMLR, 2020.
> [2] Espinosa Zarlenga, Mateo, et al. "Concept embedding models: Beyond the accuracy-explainability trade-off." Advances in Neural Information Processing Systems 35 (2022): 21400-21413.
> [3] Yuksekgonul, Mert, Maggie Wang, and James Zou. "Post-hoc concept bottleneck models." arXiv preprint arXiv:2205.15480 (2022).
> [4] Kim, Eunji, et al. "Probabilistic concept bottleneck models." arXiv preprint arXiv:2306.01574 (2023).
>
> ## Clarifying Notation
>
> We thank the reviewer for this suggestion, and we have updated Section 6 with further clarified notation and a further discussion of our derivation to ensure clarity. For example, we now clearly define $M\_{j,q,i,r}$ as the conditional probability of concept $j$ having value $q$ given that concept $i$ has value $r$.

---

> > ### Comment · Reviewer_6Tsj · 2025-05-28
> >
> > Thank the authors for their response. After reading the authors' response, several issues remain unclear:
> >
> > The authors emphasize: "Our primary contribution in this work is a new set of metrics and an analysis of locality using these metrics. While we cannot use other metrics to assess our metric (because our contribution in the paper is a set of new metrics), we do assess this new metric across a variety of concept-based models (e.g., ProbCBMs and label-free CBMs)." However, this explanation does not directly address why comparisons with other evaluation methods were not conducted. If the core contribution is "a new set of metrics and an analysis of locality using these metrics," then its validity should be demonstrated from multiple perspectives. This includes clarifying:
> >
> > 1. How the proposed metrics differ from and improve upon existing evaluation frameworks.
> >
> > 2. Whether existing interpretability methods can be assessed using these new metrics. Why can the proposed metrics improve such deployments through better-understood interpretability algorithms?
> >
> > 3. What specific characteristics of interpretability methods the proposed metrics can effectively capture.
> >
> > The experimental design in the paper does not sufficiently validate the effectiveness of the proposed metrics. As the authors mention: "We evaluate a set of common CBM models across standard CBM datasets. We focus on classification tasks because those are the most common in the CBM literature; however, even within classification, we consider both binary and multiclass classification." However, the experimental results involve very few datasets and concept-based interpretability methods, and the relationship between the interpretability methods and the proposed metrics remains unclear. We recommend that the authors include more visualizations and in-depth analyses to highlight the significance and utility of the proposed metrics for concept-based interpretability methods.
> >
> > Besides, the authors emphasize that they "analyze the robustness of concept predictors." Although Figures 4 and 5 in the experiments mention robustness, it remains unclear how this robustness is actually demonstrated in the experiments.

---

> > > ### Author Response · Authors · 2025-05-28
> > > **Author Response #1**
> > >
> > > Dear Reviewer 6Tsj,
> > >
> > > We thank you for your response, and are glad that you took the time to read through our rebuttal. We have posted answers to your questions below, and are happy to answer any additional questions you might have:
> > >
> > > ## Proposed Metric and Difference from Existing Frameworks
> > > We thank the reviewer for bringing up this question. Our primary aim in the paper is to develop a set of metrics and to study locality in concept-based models. This phenomenon had been little studied before, and so our metrics were the first to introduce and study this phenomenon. Moreover, we demonstrate that existing metrics within the concept-based space are insufficient to capture this phenomena; in Appendix B.7, we compare our metric to the OIS and NIS metric from [1], as this is the closest work in terms of proposing metrics for concept-based models. We demonstrate that the OIS and NIS metrics are unable to distinguish between models that exhibit different levels of locality leakage, implying that our new metrics are necessary to help understand locality. Finally, we emphasize that our study is among the first to study locality in concept-based models, and so we differ from existing frameworks in our emphasis and investigation into locality.
> > >
> > > [1] Zarlenga, Mateo Espinosa, et al. "Towards robust metrics for concept representation evaluation." Proceedings of the AAAI Conference on Artificial Intelligence. Vol. 37. No. 10. 2023.
> > >
> > > ## How can these Metrics improve Deployments
> > > Using our metrics, practitioners in the concept-based model space can better understand whether their models exhibit high levels of locality leakage. By better understanding this, practitioners can better understand whether their concept predictions are robust to perturbations of "irrelevant" inputs, and is a sanity check that their models perform as expected. In essence, practitioners can better determine whether their concept-based models truly reflect the presence of the underlying concepts.
> > >
> > > ## What Characteristics can these Metrics Capture
> > > Our aim is to better understand locality in concept-based models in order to understand whether the explanations made by concept-based models truly reflect the presence of the underlying concepts. As such, our methods capture the performance of the concept predictors and whether the predictions made by these concept predictors can be relied upon to understand model predictions. Models that exhibit high levels of locality leakage imply that their concept predictors are brittle and cannot be trusted, as they fail to truly reflect the presence of an underlying concept.
> > >
> > > ## Thoroughness of Experimental Design
> > > We thank the reviewer for bringing up our experimental design. In our experiments, we compare against a set of commonly used concept-based datasets (CUB and COCO) and have extensive experiments against a set of synthetic datasets. Our aim with this was to explain how and why concept-based models might fail to reflect the localities present in datasets, and we showed that this occurs in two popular concept-based datasets as well. Moreover, we evaluate against a large set of concept-based interpretability methods, including CBMs [1], CEMs [2], ProbCBMs [3], and Label-Free CBMs [4], which captures a diverse set of popular concept-based models. We thank the author for suggesting that we include more visualizations, and we note that in our experiments, we visualize the impact of various factors including model and dataset construction, though we would be happy to include any additional visualizations that might assist with understanding our results.
> > >
> > > [1] Koh, Pang Wei, et al. "Concept bottleneck models." International conference on machine learning. PMLR, 2020.
> > >
> > > [2] Espinosa Zarlenga, Mateo, et al. "Concept embedding models: Beyond the accuracy-explainability trade-off." Advances in Neural Information Processing Systems 35 (2022): 21400-21413.
> > >
> > > [3] Kim, Eunji, et al. "Probabilistic concept bottleneck models." arXiv preprint arXiv:2306.01574 (2023).
> > >
> > > [4] Oikarinen, Tuomas, et al. "Label-free concept bottleneck models." arXiv preprint arXiv:2304.06129 (2023).

---

> > > ### Author Response · Authors · 2025-05-28
> > > **Author Response #2**
> > >
> > > ## Connection to Robustness
> > > We thank the reviewer for bringing up the connection between our work and robustness. In our work, we aimed to understand whether concept predictors are robust to perturbations outside of their local region, that is, to irrelevant features. Our metrics capture different notions of perturbations, and we use these different metrics to see how robust our models are when making such perturbations. Through our experiments, we use the aforementioned metrics to understand the robustness of models; poor performance on our metrics indicates that concept-based models lack robustness, as small perturbations can drastically change the concept predictions. Our experiments show that across models and datasets, concept-based models tend to lack robust concept predictors, and so concept predictors cannot always clearly distinguish distinct concepts. At a high level, our work can be seen as an evaluation and validation of the robustness of concept predictors to different notions of perturbations.

---

### Review · Reviewer_Bbsu · 2025-04-04

**Summary Of Contributions:**

The paper investigates whether concept-based models, which explain their predictions using human-understandable intermediaries (or "concepts"), actually base these predictions on the intended, relevant features.
The authors make three key contributions: i) they introduce three novel metrics—locality leakage, locality intervention, and locality masking—to quantitatively assess if and how much a model’s concept predictions change when features outside the concept’s relevant region are perturbed, ii) they conduct an experimental study on both synthetic and real-world datasets (including CUB and COCO) to evaluate how factors like model depth, architectural choices, training procedures, and dataset diversity affect these locality metrics, and iii) they provide theoretical insights that explain how correlations between concepts in a dataset can lead models to depend on only a subset of features, even when this does not align with the true concept definition.

**Audience:**

Yes

**Claims And Evidence:**

Yes

**Requested Changes:**

- Explicitly define all variables and conditions. For example, clarify that the “perfect accuracy” assumption for the subset {$\gamma_1,\dots,\gamma_m$} means that for those concepts, the predictor has zero error.
- Clearly state the domain and range of each variable and what the probability terms represent.

In addition, the following discussion would improve the manuscript and strengthen their contributions, if possible (but not necessary).

### Discuss tightness and optimality of the bound
- Elaborate on whether the bound $\epsilon \leq \alpha + (1 − \beta)^s$ is tight under realistic conditions or if it can be improved further.
- Consider discussing conditions or examples where the bound becomes loose or where additional constraints (e.g., independence assumptions) could yield a sharper bound.

### Relax the “perfect accuracy” requirement
- Investigate how the theorem might extend if the known concepts are predicted with high—but not perfect—accuracy.
- For example, include an error term $\delta$ for the prediction of the known concepts and derive an adjusted bound

**Strengths And Weaknesses:**

- The paper introduces three novel and well-motivated metrics—locality leakage, locality intervention, and locality masking—that provide a way to evaluate whether concept-based models base their predictions on the intended, relevant features.
- It features an extensive experimental evaluation spanning synthetic and real-world datasets (such as CUB and COCO). This thorough evaluation covers different architectural choices (e.g., model depth, receptive field, parameter count) and training modifications (like early stopping, adversarial training, and regularization).
- The inclusion of theoretical insights supports the empirical findings and offers a deeper understanding of how dataset correlations can lead to models relying on spurious features.
- The manuscript offers practical guidance by recommending that practitioners use diverse datasets and models with appropriate capacity (e.g., smaller models or those with improved representational power) to obtain more trustworthy concept explanations.

Overall, the work addresses a timely and important challenge in explainable AI by questioning the reliability of concept-based explanations, which is critical for model trustworthiness.

##

---

> ### Author Response · Authors · 2025-05-05
> **Author Response**
>
> Dear Reviewer Bbsu,
>
> We thank you for your comments and suggestions. Moreover, we are glad you find our methods “well-motivated” and our experimentation “extensive”. We have updated our manuscript based on your suggestions and have marked any changes made in red. We provide answers to your questions below:
>
> ## Explicitly Define Variables and Domains
>
> We thank the reviewer for this suggestion and have updated our description accordingly in Section 6. In particular, we have specified domains for $q$,$r$,$M\_{j,q,i,r}$, and $j$, and have further specified what "perfect accuracy" and "using only information about the concepts.." means.
>
> ## Realistic Bound
>
> We thank the reviewer for their interest in understanding the tightness of the bound. We include a sketch of the bound’s tightness and provide this information in Section 6 of our updated manuscript.
>
> To make the tightness of the bound more explicit, we can write it as follows: $(1-(1-\\beta)^{s}) \\alpha + \\frac{1}{2}*(1-\\beta)^{s}$. Here, $(1-\\beta)^{s}$ represents the probability that one of the $s$ matching triplets holds; if one of these holds, then we make a mistake with probability $\\alpha$. If none of the $s$ triplets holds, then we simply pick the majority label, which fails with probability at most $\\frac{1}{2}$. Our current bound for $\\epsilon$ is a simplified, but looser, upper bound on this.
>
> In light of this, our bound is an approximation because we assume that if none of the $s$ triplets holds, then predicting the majority label holds with probability $\\frac{1}{2}$, and because we assume $M\_{j,q,i,r} \\geq 1-\\alpha$ rather than $M\_{j,q,i,r} = 1-\\alpha$ (and similarly for $\\mathbb{P}[\\mathbf{c}\_{i}=r] \\geq \\beta$). Our bound holds exactly if $M\_{j,q,i,r} = 1-\\alpha$, $\\mathbb{P}[\\mathbf{c}\_{i}=r] = \\beta$, and concept $j$ is evenly balanced between 0 or 1. Alternatively, as $(1-\\beta)^{s}$ becomes small, the $\\alpha$ term dominates our bound, and our bound becomes tighter. Intuitively, this holds when $s$ is large (say there are many correlated concepts), or $\\beta$ is near $1$ (when $\\mathbb{P}[\\mathbf{c}\_{i}=r] = 1$).
>
> ## Extension to Imperfect Concepts
>
> We thank the reviewer for asking about an extension to the imperfect concept predictor scenario. In this scenario, we identify three sources of errors:
> With probability $(1-(1-\\beta)^{s}) \\delta$, the concept $i$ is incorrectly predicted;
> with probability $(1-(1-\\beta)^{s}) (1-\\delta)\\alpha$, $\\mathbb{P}[\\mathbf{c}\_{j}=q | \\mathbf{c}\_{i}=r]$ is incorrect (that is $\\mathbf{c}\_{i}=r$, but $\\mathbf{c}\_{j} \\neq q$);
> And, with probability (1-\\beta)^{s}, no such triplet is found.
> We can then bound $\\epsilon$ as $(1-(1-\\beta)^{s}) (1-\\delta) \\alpha + (1-(1-\\beta)^{s})\\delta + \\frac{1}{2}(1-\\beta)^{s}$, which we can further upper bound as $\\alpha + \\delta + (1-\\beta)^{s}$. Essentially, with errors of size $\\delta$, we loosen our bound by $\\delta$. We have incorporated this additional extension into Section 6 of our manuscript.

---

### Review · Reviewer_APxv · 2025-04-25

**Summary Of Contributions:**

This work proposes a new perspective on assessing the faithfulness of concept predictors in the context of concept bottleneck models, named "locality", which evaluates whether concept predictors rely only on relevant regions in the input image. It proposes three metrics to quantify locality -- locality leakage, locality intervention, and locality masking. They demonstrate these metrics in various experiments and provide insights on how correlations of concepts affect locality.

**Audience:**

Yes

**Broader Impact Concerns:**

There are no concerns about the ethical implications of the work.

**Claims And Evidence:**

Yes

**Requested Changes:**

1. Adjustments that are critical to secure my recommendation for acceptance.
- 1.1 Clarify the details concerning the methods as listed in the weakness part.
- 1.2 Justify or adjust the experiment settings listed in the weakness part.
2. Adjustment that will further strengthen the work, in my view
- 2.1 Discuss and compare other metrics for assessing concept faithfulness.
- 2.2  Make a connection between locality and intervention by discussing how locality affects the efficiency of the model intervention.

**Strengths And Weaknesses:**

**Strengths**
1. The faithfulness of the concept predictor plays an important role in concept-based explanations, and this work provides a valuable, new way to assess it.
2. The method is solid, and the presentation is generally clear.
3. The experiments on different model architectures, training strategies, and hyperparameters are comprehensive.

**Weakness**
1. Some details concerning methods are missing.
- 1.1 When measuring the locality leakage, how are adversarial perturbations conducted?
- 1.2 When measuring the locality intervention, how to make sure the perturbations do not alter the concept labels?
2. Some experimental settings need further clarification and adjustment.
- 2.1 In the synthetic experiment, it’s not clear to me why there are k concepts corresponding to k/2 regions. It seems to imply that concepts are region-specific and enforce the concept predictor to only focus on a designated region (left or right in the 2-object case). In the general image classification set, objects concerning a certain concept are usually not present in a designated region.
- 2.2 In the locality masking experiments on CUB, I’m not sure the mask with a radius of 0.3l is reasonable for all the concepts. Different concepts intrinsically have different sizes (e.g., belly vs. beak). The bird region also has various sizes in different images. This rough masking strategy might mask relevant (irrelevant) regions unintentionally.
3. Lack of discussion and comparison with other methods
- 3.1 What are other methods/metrics assessing the concept of faithfulness, and how does locality compare with these metrics

---

> ### Author Response · Authors · 2025-05-05
> **Author Response**
>
> Dear Reviewer APxv,
>
> We thank you for your helpful comments and appreciate that you find our work “comprehensive” and “valuable”. We have updated our manuscript based on your suggestions and have marked any changes made in red. We provide answers to your questions below:
>
> ## Constructing Adversarial Perturbations
>
> To construct adversarial perturbations for our locality leakage metric, we optimize for an input $\mathbf{x}^\prime$ that does not modify features in its local region $\mathcal{L}(\mathbf{x}^\prime)$, while maximally modifying the concept prediction $g(\mathbf{x}^\prime)_{j}$. We do this through projected gradient descent, using such a method to optimize over feasible $\mathbf{x}^\prime$ given a sample $\mathbf{x}$ [1]. We have included this information in an updated version of Section 5.2.
>
> [1] Madry, Aleksander, et al. "Towards Deep Learning Models Resistant to Adversarial Attacks." International Conference on Learning Representations. 2018.
>
> ## Locality Intervention and Inter-Concept Alteration
>
> We thank the reviewer for bringing up this question. We ensure that the perturbations for locality intervention do not impact the true concept label by constructing a dataset with many different combinations of concepts. We can do so in a synthetic controlled environment to simulate different combinations of concepts accordingly. We can then precompute the impact of perturbing a concept by storing data points for different concept combinations, as this ensures that perturbations do not impact concept labels. More generally, we ensure that we do not change concepts during perturbation because we rely on alternate data points found in our dataset with the specified concept value, which ensures that the concept label remains unchanged. We include these details in Section 5.3 of our updated manuscript.
>
> ## Number of regions in the Synthetic Dataset
>
> In our synthetic dataset, we have $k/2$ spatial regions (e.g., left and right positions), where each region can contain either a “triangle” or a “square”. Each region is associated with two binary concepts: "is triangle present" and "is square present." This gives us $k$ concepts total ($2$ concepts $\times$ $k/2$ regions). For example, in a 2-region setup (left and right), we have 4 concepts: "left triangle present," "left square present," "right triangle present," and "right square present." For any given image, exactly one concept is active for each region (either triangle or square is present), making the concept structure localized to specific regions by design. This controlled setup allows us to precisely evaluate whether concept predictors correctly identify this region-specific information.
>
> In the synthetic dataset, we agree that each concept is localised to a specific region. We construct the synthetic dataset in such a way that we can easily assess the locality of concept predictors. In real-world datasets, while the exact region that a concept is present in might change, many concepts are still localised to specific regions. For example, information on a bird's “head color” is only found in a specific region of the image. So while the exact region might change, the property of locality remains constant.
>
> ## Mask Size on CUB
>
> We thank the reviewer for asking an interesting question about masking on the CUB dataset. We use masking with a set radius because the CUB dataset does not give the exact bounding box for bird parts, but rather gives centres for different body parts. In our updated manuscript in Section 5.3 and Appendix B.6, we include more details on how our masking procedure works, and include these below:
> We only mask features that are closer to one body than another. For example, when masking the beak of a bird, if a pixel is closer to the head than the beak, then we do not mask it. We do so to naturally capture the idea that we should mask irrelevant regions, and this serves as a heuristic to avoid that.
> In Appendix B.6, we experiment with different mask sizes. We find that, although the selection of mask radius does impact concept predictions, our overall takeaway remains: regardless of the mask size, the impact of relevant masking is small.

---

> ### Author Response · Authors · 2025-05-05
> **Author Response #2**
>
> ## Comparison to Other Metrics
>
> We thank the reviewer for bringing up the suggestion to compare with other metrics. We will first contrast our work with [1], which proposes oracle and niche impurity scores (OIS and NIS, respectively) to measure the level of impurity or leakage contained within concept predictions. While both our metric and theirs are concerned with analyzing the effectiveness of concept-based models, our metric is more concerned with the features leveraged by concept predictors, while [1] is more concerned about leaking information into concept predictions. Moreover, in Appendix B.7, we include a comparison with the OIS and NIS metrics, where we demonstrate that the OIS and NIS metrics cannot distinguish between predictors that exhibit differing levels of locality leakage. In essence, the metrics proposed in [1] complement our metrics and measure different desiderata for concept predictors.
>
> We also contrast our work with [2], which focuses on shortcut learning in Neuro-symbolic settings. Both our work and [2] are focused on understanding shortcuts in interpretable models. However, we differ in the setting; we focus on the concept-based learning setting, while [2] focuses on Neuro-symbolic models. Moreover, [2] focuses on shortcut learning. In contrast, we focus on understanding concept-feature relationships, which might cause concept predictors to learn shortcuts, but not necessarily always (e.g., there are no shortcuts to learn in the synthetic dataset). We include a discussion of these two related works in an updated Section 2.
>
> Besides these external metrics, other notions of concept faithfulness include standard metrics such as concept accuracy or mean concept AUC-ROC. These metrics, however, are interested in measuring the faithfulness of concept predictions rather than whether they are faithfully capturing the appropriate concept (meaning they could not discriminate between a model exploiting spurious correlations to predict a concept and one that does not exploit those spurious correlations).
>
> [1] Zarlenga, Mateo Espinosa, et al. "Towards robust metrics for concept representation evaluation." Proceedings of the AAAI Conference on Artificial Intelligence. Vol. 37. No. 10. 2023.
>
>
> [2] Marconato, Emanuele, et al. "Not all neuro-symbolic concepts are created equal: Analysis and mitigation of reasoning shortcuts." Advances in Neural Information Processing Systems 36 (2023): 72507-72539.
>
> ## Relationship between Locality and Intervention
>
> Models that better respect locality can also improve performance on intervention tasks because concept predictors leverage relevant features, which could improve performance and require fewer interventions for perfect performance. At the same time, locality is largely concerned with learning better concept predictors, while intervention is more concerned with label predictors accurately responding to changes in concepts. From this perspective, locality and intervention are complementary; intervention fixes existing issues in concept predictors, while locality points to the reasons why these concept predictors might not perform well.

---

### Decision · Action_Editor_fnk6 · 2025-06-16

**Recommendation:** Accept as is

**Audience:**

Yes

**Audience Explanation:**

The interpretability of complex and modern ML models is matter of significance interest to a large part of the TMLR community.

**Claims And Evidence:**

Yes

**Claims Explanation:**

This paper aims at clarifying some misconceptions of concept bottleneck models for interpretability in machine learning. In particular, the authors study aspects of locality of concept-based explanations, focusing on whether concept predictors rely on relevant input features rather than spurious correlations. The paper introduces three metrics to make this assessment: locality leakage, intervention, and masking. The authors demonstrate theoretically why correlations between concepts can manifests as violations of locality, and provide extensive experiments on synthetic and real datasets across different model architectures.

The review process was mostly productive. Two reviewers found the paper valuable to the TMLR community, and provided constructive comments that further improved the paper. A third reviewer (6Ts) was significantly less supportive, and had severe concerns regarding (i) clarity and presentation, (ii) limited comparison with other methods and metrics, (iii) shallow experimental validation, and (iv) theoretical assumptions. After evaluating the authors responses, I believe (i) to be mostly not well supported, and (ii-iv) have been addressed by the authors. Lastly, and unfortunately, reviewer 6Ts never provided a recommendation.

I am therefore recommending this paper for acceptance, while stressing that the authors should revise all added content that addresses the reviewers' comments for preciseness and correctness (e.g. there are some typos in the added text, "Each of
our metrics capture[s]..", some missing periods, etc). Please revise and adjust in the camera ready version.